# AmbientFlow: Invertible generative models from incomplete, noisy measurements

**Varun A. Kelkar**  *varun.kelkar@analog.com*
*University of Illinois at Urbana-Champaign, Urbana, IL 61801*
*Currently at Analog Devices, Inc., Boston, MA 02110*

**Rucha M. Deshpande**  *r.deshpande@wustl.edu*
*Washington University in St. Louis, St. Louis, MO 63130*

**Arindam Banerjee**  *arindamb@illinois.edu*
*University of Illinois at Urbana-Champaign, Urbana, IL 61801*

**Mark A. Anastasio**  *maa@illinois.edu*
*University of Illinois at Urbana-Champaign, Urbana, IL 61801*

**Reviewed on OpenReview:** *https://openreview.net/forum?id=txpYITR8oa*

## Abstract

Generative models have gained popularity for their potential applications in imaging science, such as image reconstruction, posterior sampling and data sharing. Flow-based generative models are particularly attractive due to their ability to tractably provide exact density estimates along with fast, inexpensive and diverse samples. Training such models, however, requires a large, high quality dataset of objects. In applications such as computed imaging, it is often difficult to acquire such data due to requirements such as long acquisition time or high radiation dose, while acquiring noisy or partially observed measurements of these objects is more feasible. In this work, we propose AmbientFlow, a framework for learning flow-based generative models directly from noisy and incomplete data. Using variational Bayesian methods, a novel framework for establishing flow-based generative models from noisy, incomplete data is proposed. Extensive numerical studies demonstrate the effectiveness of AmbientFlow in learning the object distribution. The utility of AmbientFlow in a downstream inference task of image reconstruction is demonstrated.

## 1 Introduction

Generative models have become a prominent focus of machine learning research in recent years. Modern generative models are neural network-based models of unknown data distributions learned from a large number of samples drawn from the distribution. They ideally provide an accurate representation of the distribution of interest and enable efficient, high-quality sampling and inference. Most modern generative models are implicit, i.e. they map a sample from a simple, tractable distribution such as the standard normal, to a sample from the distribution of interest. Popular generative models for image data distributions include variational autoencoders (VAEs), generative adversarial networks (GANs), normalizing flows and diffusion probabilistic models, among others (Kingma *et al.*, 2019; Goodfellow *et al.*, 2020; Kingma & Dhariwal, 2018; Ho *et al.*, 2020). Recently, many of these models have been successful in synthesizing high-quality perceptually realistic images from the underlying distribution.

Generative models have also been investigated for applications in imaging science. For example, computed imaging systems such as computed tomography (CT) or magnetic resonance imaging (MRI) rely on computational reconstruction to obtain an estimate of an object from noisy or incomplete imaging measurements. Generative models have been investigated for their use as priors in image reconstruction in order to mitigate

the effects of data-incompleteness or noise in the measurements. While GANs have been explored for this purpose (Bora *et al.*, 2017; Menon *et al.*, 2020; Kelkar & Anastasio, 2021), they suffer from shortcomings such as inadequate mode coverage (Thanh-Tung & Tran, 2020), insufficient representation capacity (Karras *et al.*, 2020; Bora *et al.*, 2017) and misrepresentation of important domain-specific statistics (Kelkar *et al.*, 2023a). On the other hand, invertible generative models (IGMs), or normalizing flows offer exact density estimates, tractable log-likelihoods and useful representations of individual images (Dinh *et al.*, 2016; 2014; Kingma & Dhariwal, 2018; Kelkar *et al.*, 2021), making them more reliable for downstream inference tasks in imaging science (Asim *et al.*, 2020; Jalal *et al.*, 2021a). These models have shown potential for use in tasks such as image reconstruction, posterior sampling, uncertainty quantification and anomaly detection (Kelkar *et al.*, 2021; Asim *et al.*, 2020; Khorashadizadeh *et al.*, 2023; Jalal *et al.*, 2021a; Zhao *et al.*).

Although IGMs are attractive for imaging applications, training them requires a large dataset of objects or high-quality image estimates as a proxy for the objects. Building such a dataset is challenging since acquiring a complete set of measurements to uniquely specify each object can be infeasible. Therefore, it is of interest to learn an IGM of objects directly from a dataset of noisy, incomplete imaging measurements. This problem was previously addressed in the context of GANs via the AmbientGAN framework, which augments a conventional GAN with the measurement operator (Bora *et al.*, 2018; Zhou *et al.*, 2022). It consists of generator and discriminator networks that are jointly optimized via an adversarial training strategy. Here, the generator attempts to synthesize synthetic objects that produce realistic measurements. The real and synthetic object distributions are then compared *indirectly* via a discriminator that distinguishes between real measurements and simulated measurements of the synthetic objects. This is fundamentally different from the training procedure of IGMs, which *directly* computes and maximizes the log-probability of the training data samples. Therefore, ideas from the AmbientGAN framework cannot be easily adapted to train an IGM of objects when only incomplete measurements is available.

The key contributions of this work are as follows. First a new framework named AmbientFlow is developed for training IGMs using noisy, incomplete measurements. Second, the accuracy of the object distribution recovered via AmbientFlow is theoretically analyzed under prescribed ideal conditions using compressed sensing. Next, numerical studies are presented to demonstrate the effectiveness of the proposed method on several different datasets and measurement operators. Finally, the utility of AmbientFlow for a downstream Bayesian inference task is illustrated using a case study of image reconstruction from simulated stylized MRI measurements.

The remainder of this manuscript is organized as follows. Section 2 describes the background of invertible generative models, computed imaging systems and image reconstruction from incomplete measurements. Section 3 describes the notation used and the proposed approach. Section 4 describes the setup for the numerical studies, with the results being presented in Section 5. Finally, the discussion and conclusion is presented in Section 6.

## 2 Background

**Invertible generative models.** Invertible generative models (IGMs) or flow-based generative models, are a class of generative models that employ an invertible neural network (INN) to learn an implicit mapping from a simple distribution such as an independent and identically distributed (iid) Gaussian distribution to the data distribution of interest. The INN is a bijective mapping $G_\theta : \mathbb{R}^n \to \mathbb{R}^n$ constructed using a composition of $L$ simpler bijective functions $g_i : \mathbb{R}^n \to \mathbb{R}^n$, with

$$\mathbf{h}^{(i)} = g_i(\mathbf{h}^{(i-1)}) = (g_i \circ g_{i-1} \circ \cdots \circ g_1)(\mathbf{z}), \quad 0 < i \leq L, \quad \mathbf{z}, \mathbf{h}^{(i)} \in \mathbb{R}^n, \tag{1}$$

and $\mathbf{x} = G_\theta(\mathbf{z}) = \mathbf{h}^{(L)}$. As a consequence of bijectivity, the probability distribution function (PDF) $p_\theta$ of $\mathbf{x}$ represented by the IGM can be related to the PDF $q_{\mathbf{z}}$ of $\mathbf{z}$ as:

$$p_\theta(\mathbf{x}) \cdot | \det \nabla_{\mathbf{z}} G_\theta(\mathbf{z}) | = q_{\mathbf{z}}(\mathbf{z}). \tag{2}$$

In order the establish the IGM, a dataset $\mathcal{D} = \{\mathbf{x}^{(i)}\}_{i=1}^D$ is used, where $\mathbf{x}^{(i)}$ are assumed to be independent draws from the unknown true distribution $q_{\mathbf{x}}$. The INN is then trained by minimizing the following negative

log-likelihood objective over the training dataset $\mathcal{D}$:

$$\mathcal{L}(\theta) = -\sum_{i=1}^{D} \log p_\theta(\mathbf{x}^{(i)}) = \sum_{i=1}^{D} \left[ \log q_\mathbf{z}(\mathbf{z}^{(i)}) - \log |\det \nabla_\mathbf{z} G_\theta(\mathbf{z}^{(i)})| \right], \quad \mathbf{z}^{(i)} = G_\theta^{-1}(\mathbf{x}^{(i)}). \tag{3}$$

Minimizing the above loss function is equivalent to minimizing the Kullback–Leibler (KL) divergence $D_{\mathrm{KL}}(q_\mathbf{x} \| p_\theta) := \mathbb{E}_{\mathbf{x} \sim q_\mathbf{x}}[\log q_\mathbf{x}(\mathbf{x}) - \log p_\theta(\mathbf{x})]$ between $p_\theta$ and the true data distribution $q_\mathbf{x}$. Most invertible generative models utilize a specialized architecture in order to guarantee invertibility of the constituent functions $g_i$ and tractable computation of the Jacobian loss term $\log |\det \nabla_\mathbf{z} G_\theta(\mathbf{z})|$. Popular building blocks of such an architecture are affine coupling layers (Dinh *et al.*, 2016), random permutations (Dinh *et al.*, 2014) and invertible $1 \times 1$ convolutions (Kingma & Dhariwal, 2018).

**Image reconstruction from incomplete imaging measurements.** Many computed imaging systems can be modeled using an imaging equation described by the following linear system of equations:

$$\mathbf{g} = H\tilde{\mathbf{f}} + \mathbf{n}, \tag{4}$$

where $\tilde{\mathbf{f}} \in \mathbb{R}^n$ is approximates the object to-be-imaged, $H \in \mathbb{R}^{m \times n}$, known as the forward model, is a linear operator that models the physics of the imaging process, $\mathbf{n} \in \mathbb{R}^m, \mathbf{n} \sim q_\mathbf{n}$ models the measurement noise, and $\mathbf{g} \in \mathbb{R}^m$ are the measurements of the object. Often, $H$ is ill-conditioned or rank deficient, in which case the measurements $\mathbf{g}$ are not sufficient to form a unique and stable estimate $\hat{\mathbf{f}}$ of the object $\tilde{\mathbf{f}}$, and prior knowledge about the nature of $\tilde{\mathbf{f}}$ is needed. Traditionally, one way to incorporate this prior information is to constrain the domain of $H$. For example, compressed sensing stipulates that if the true object is $k$-sparse after a full-rank linear transformation $\Phi \in \mathbb{R}^{l \times n}$, $l \geq n$, then the object can be stably estimated if for all vectors $\mathbf{v} \in \mathbb{R}^n$ that are $k$-sparse in the transform domain $\Phi$, $H$ satisfies the restricted isometry property (RIP) (Candes *et al.*, 2006):

**Definition 2.1** (Restricted isometry property). For $s \in \mathbb{N}$, define the restricted isometry constant (RIC) $\delta_s$ as the smallest constant that satisfies

$$(1 - \delta_s)\|\mathbf{v}\|_2^2 \leq \|H\mathbf{v}\|_2^2 \leq (1 + \delta_s)\|\mathbf{v}\|_2^2, \tag{5}$$

for all $\mathbf{v}$ such that $\|\Phi\mathbf{v}\|_0 \leq s$. $H$ is said to satisfy the restricted isometry property for all $\mathbf{v}$ such that $\|\Phi\mathbf{v}\|_0 \leq k$, if $\delta_k + \delta_{2k} + \delta_{3k} < 1$ (Candes *et al.*, 2006).

## 3  Approach

In this section, an AmbientFlow method is proposed for obtaining an IGM of objects from a dataset of measurements. The following preliminary notation will be used in the remainder of this paper.

**Notation.** Let $q_\mathbf{f}$, $q_\mathbf{g}$ and $q_\mathbf{n}$ denote the unknown true object distribution to-be-recovered, the true measurement distribution and the known measurement noise distribution, respectively. Let $\mathcal{D} = \{\mathbf{g}^{(i)}\}_{i=1}^D$ be a dataset of independent and identically distributed (iid) measurements drawn from $q_\mathbf{g}$. Let $G_\theta : \mathbb{R}^n \to \mathbb{R}^n$ be an INN. Let $p_\theta$ be the distribution represented by $G_\theta$, i.e. given a latent distribution $q_\mathbf{z} = \mathcal{N}(\mathbf{0}, I_n)$, $G_\theta(\mathbf{z}) \sim p_\theta$ for $\mathbf{z} \sim q_\mathbf{z}$. Also, let $\psi_\theta$ be the distribution of synthetic measurements, i.e. for $\mathbf{f} \sim p_\theta$, $H\mathbf{f} + \mathbf{n} \sim \psi_\theta$. Let $p_\theta(\mathbf{f} \,|\, \mathbf{g}) \propto q_\mathbf{n}(\mathbf{g} - H\mathbf{f}) \, p_\theta(\mathbf{f})$ denote the posterior induced by the learned object distribution represented by $G_\theta$. Let $\Phi \in \mathbb{R}^{l \times n}$, $l \geq n$ be a full-rank linear transformation (henceforth referred to as a sparsifying transform). Also, let $\mathcal{S}_k = \{\mathbf{v} \in \mathbb{R}^n \text{ s.t. } \|\Phi\mathbf{v}\|_0 \leq k\}$ be the set of vectors $k$-sparse with respect to $\Phi$. Since $\Phi$ is full-rank, throughout this work we assume without the loss of generality, that $\|\Phi^+\|_2 \leq 1$, where $\Phi^+$ is the Moore-Penrose pseudoinverse of $\Phi$. Throughout the manuscript, we also assume that $q_\mathbf{f}$ is absolutely continuous with respect to $p_\theta$, and $q_\mathbf{g}$ is absolutely continuous with respect to $\psi_\theta$.

Conventionally, according to the discussion below Eq. (3), $D_{\mathrm{KL}}(q_\mathbf{f} \| p_\theta)$ would have to be minimized in order to train the IGM to estimate $q_\mathbf{f}$. However, in the present scenario, only samples from $q_\mathbf{g}$ are available. Therefore, we attempt to minimize the divergence $D_{\mathrm{KL}}(q_\mathbf{g} \| \psi_\theta)$, and show that for certain scenarios of interest, this

is formally equivalent to approximately minimizing a distance between $q_{\mathbf{f}}$ and $p_\theta$. However, computing $D_{\mathrm{KL}}(q_{\mathbf{g}}\|\psi_\theta)$ is non-trivial because a direct representation of $\psi_\theta(\mathbf{g})$ that could enable the computation of $\log \psi_\theta(\mathbf{g})$ is not available. Fortunately, a direct representation of $p_\theta$ is available via $G_\theta$, which can be used to compute $\log p_\theta(\mathbf{f})$ for a given $\mathbf{f} \in \mathbb{R}^n$, using Eq. (2). Therefore, an additional INN, known as the posterior network $h_\phi(\,\cdot\,;\mathbf{g}) : \mathbb{R}^n \to \mathbb{R}^n$ is introduced that represents the model posterior $p_\phi(\mathbf{f}\,|\,\mathbf{g})$ designed to approximate $p_\theta(\mathbf{f}\,|\,\mathbf{g})$ when jointly trained along with $G_\theta$. The posterior network $h_\phi$ is designed to take two inputs – a new latent vector $\boldsymbol{\zeta} \sim q_{\boldsymbol{\zeta}} = \mathcal{N}(\mathbf{0}, I_n)$, and an auxiliary conditioning input $\mathbf{g} \sim q_{\mathbf{g}}$ from the training dataset, to produce $h_\phi(\boldsymbol{\zeta};\mathbf{g}) \sim p_\phi(\mathbf{f}\,|\,\mathbf{g})$. The following theorem establishes a loss function that minimizes $D_{\mathrm{KL}}(q_{\mathbf{g}}\|\psi_\theta)$ using the posterior network, circumventing the need for direct access to $\psi_\theta(\mathbf{g})$, or samples of true objects from $q_{\mathbf{f}}$.

**Theorem 3.1.** *Let $h_\phi$ be such that $p_\phi(\mathbf{f}\,|\,\mathbf{g}) > 0$ over $\mathbb{R}^n$. Minimizing $D_{\mathrm{KL}}(q_{\mathbf{g}}\|\psi_\theta)$ is equivalent to maximizing the objective function $\mathcal{L}(\theta, \phi)$ over $\theta, \phi$, where $\mathcal{L}(\theta, \phi)$ is defined as*

$$\mathcal{L}(\theta, \phi) = \mathbb{E}_{\mathbf{g} \sim q_{\mathbf{g}}} \left[ \log \mathbb{E}_{\boldsymbol{\zeta} \sim q_{\boldsymbol{\zeta}}} \left\{ \frac{p_\theta\big(h_\phi(\boldsymbol{\zeta};\mathbf{g})\big)\ q_{\mathbf{n}}\big(\mathbf{g} - H h_\phi(\boldsymbol{\zeta};\mathbf{g})\big)}{p_\phi\big(h_\phi(\boldsymbol{\zeta};\mathbf{g})\,|\,\mathbf{g}\big)} \right\} \right] \tag{6}$$

The proof of Theorem 3.1 is provided in the appendix. A variational lower bound of $\mathcal{L}$ is employed, which promotes consistency between the modeled posterior $p_\phi(\,\cdot\,|\,\mathbf{g})$ and the posterior induced by the learned object distribution, $p_\theta(\,\cdot\,|\,\mathbf{g})$:

$$\mathcal{L}_M(\theta, \phi) = \mathbb{E}_{\mathbf{g}, \boldsymbol{\zeta}_i} \operatorname*{logavgexp}_{0 < i \leq M} \left[ \log p_\theta\big(h_\phi(\boldsymbol{\zeta}_i;\mathbf{g})\big) + \log q_{\mathbf{n}}\big(\mathbf{g} - H h_\phi(\boldsymbol{\zeta}_i;\mathbf{g})\big) - \log p_\phi\big(h_\phi(\boldsymbol{\zeta}_i;\mathbf{g})\,|\,\mathbf{g}\big) \right], \tag{7}$$

where $\boldsymbol{\zeta}_i \sim q_{\boldsymbol{\zeta}}$, $0 < i \leq M$, and $\operatorname{logavgexp}_{0 < i \leq M}(x_i) := \log \left[ \frac{1}{M} \sum_{i=1}^{M} \exp(x_i) \right]$.

Intuitively, the three terms inside logavgexp in Eq. (7) can be interpreted as follows. The first term implies that $G_\theta$ is trained on samples produced by the posterior network $h_\phi$. The second term is a data-fidelity term that makes sure that $h_\phi$ produces objects consistent with the measurement model. The third term penalizes degenerate $h_\phi$ for which $\nabla_{\boldsymbol{\zeta}} h_\phi(\boldsymbol{\zeta};\mathbf{g})$ is ill-conditioned, for example when $h_\phi$ produces no variation due to $\boldsymbol{\zeta}$ and only depends on $\mathbf{g}$. Note that the first and third terms are directly accessible via the INNs $G_\theta$ and $h_\phi$. Also, for noise models commonly used in modeling computed imaging systems, such as the Gaussian noise model (Barrett & Myers, 2013), $q_{\mathbf{n}}$ can be explicitly computed.

For sufficiently expressive parametrizations for $p_\theta$ and $h_\phi$, the maximum possible value of $\mathcal{L}_M$ is $\mathbb{E}_{\mathbf{g} \sim q_{\mathbf{g}}} \log q_{\mathbf{g}}(\mathbf{g})$, which corresponds to the scenario where the learned posteriors are consistent, i.e. $p_\phi(\mathbf{f}\,|\,\mathbf{g}) = p_\theta(\mathbf{f}\,|\,\mathbf{g})$, and the learned distribution of measurements matches the true measurement distribution, i.e. $\psi_\theta = q_{\mathbf{g}}$. It can be shown that for a class of forward operators, matching the measurement distribution is equivalent to matching the object distribution:

**Lemma 3.1.** *If $H$ is a square matrix ($n = m$) with full-rank, if the noise $\mathbf{n}$ is independent of the object, and if the characteristic function of the noise $\chi_{\mathbf{n}}(\mathbf{v}) = \mathbb{E}_{\mathbf{n} \sim q_{\mathbf{n}}} \exp(\iota\,\mathbf{v}^\top \mathbf{n})$ has full support over $\mathbb{R}^m$ ($\iota$ is the square-root of $-1$), then $\psi_\theta = q_{\mathbf{g}} \Rightarrow p_\theta = q_{\mathbf{f}}$.*

The proof of Lemma 3.1 is provided in Appendix A. Lemma 3.1 can be extended to a certain class random, rank-deficient forward operators that nevertheless provide an invertible push-forward operator (Bora *et al.*, 2018). However, in computed imaging, forward models are often deterministic with a fixed null-space, where it is typically not possible to design the hardware to ensure the invertibility of the push-forward operator (Graff & Sidky, 2015; Lustig *et al.*, 2008). In such a setting, it is not possible to uniquely relate the learned object distribution $p_\theta$ to the learned measurement distribution $\psi_\theta$ without additional information about $q_{\mathbf{f}}$. Nevertheless, if the objects of interest are known to be compressible with respect to a sparsifying transform $\Phi$, $p_\theta$ can be constrained to the set of distributions concentrated on these compressible objects. In order to recover a distribution $p_\theta$ concentrated on objects that are compressible with respect to $\Phi$, the following optimization problem is proposed:

$$\hat{\theta}, \hat{\phi} = \arg\min_{\theta, \phi} -\mathcal{L}_M(\theta, \phi) \quad \text{subject to} \quad \mathbb{E}_{\mathbf{g} \sim q_{\mathbf{g}}} \mathbb{E}_{\mathbf{f} \sim p_\phi(\cdot\,|\,\mathbf{g})} \|\Phi \mathbf{f} - \Phi \operatorname{proj}_{\mathcal{S}_k}(\mathbf{f})\|_1 < \epsilon, \tag{8}$$

where $\mathrm{proj}_{\mathcal{S}_k}(\mathbf{f})$ denotes the orthogonal projection of $\mathbf{f} \in \mathbb{R}^n$ onto the set $\mathcal{S}_k$ of objects for which $\Phi\mathbf{f}$ is $k$−sparse. It can be shown that if $H$ and $\mathbf{f} \sim q_{\mathbf{f}}$ satisfy the conditions of compressed sensing and the AmbientFlow is trained sufficiently well using Eq. (8), then the error between the true and recovered object distributions can be bounded. This is formalized as follows.

**Theorem 3.2.** *For a PDF $q : \mathbb{R}^n \to \mathbb{R}$, let $q^{\mathcal{S}_k}$ denote the distribution of $\mathrm{proj}_{S_k}(\mathbf{x})$, for $\mathbf{x} \sim q$. Also, for distributions $q_1, q_2$, let $W_1(q_1 \| q_2) := \inf_{q \in \Gamma} \mathbb{E}_{(\mathbf{x_1}, \mathbf{x_2}) \sim q} \|\mathbf{x}_1 - \mathbf{x}_2\|_2$, denote the Wasserstein 1-distance, with $\Gamma$ being the set of all joint distributions $q : \mathbb{R}^{n \times n} \to \mathbb{R}$ with marginals $q_1, q_2$, i.e. $\int q(\mathbf{x}_1, \mathbf{x}_2) d\mathbf{x}_2 = q_1(\mathbf{x}_1), \ \int q(\mathbf{x}_1, \mathbf{x}_2) d\mathbf{x}_1 = q_2(\mathbf{x}_2)$.*

*If the following hold:*
1. *$W_1(q_{\mathbf{f}} \| q_{\mathbf{f}}^{\mathcal{S}_k}) \leq \epsilon'$ (the true object distribution is concentrated on k-sparse objects under $\Phi$),*
2. *$H$ satisfies the RIP for objects k-sparse w.r.t. $\Phi$, with isometry constant $\delta_k$,*
3. *the characteristic function of noise $\chi_{\mathbf{n}}(\mathbf{v})$ has full support over $\mathbb{C}^m$, and*
4. *$(\theta, \phi)$ satisfying $p_\theta = q_{\mathbf{f}}$ and $p_\phi(\cdot \,|\, \mathbf{g}) = p_\theta(\cdot \,|\, \mathbf{g})$ is a feasible solution to Eq. (8) ($G_\theta$ and $h_\phi$ have sufficient capacity),*

*then the distribution $p_{\hat{\theta}}$ recovered via Eq. (8) is close to the true object distribution, in terms of the Wasserstein distance i.e.*

$$W_1(p_{\hat{\theta}} \| q_{\mathbf{f}}) \leq \left(1 + \frac{1}{\sqrt{1 - \delta_k}} \|H\|_2 \right)(\epsilon + \epsilon'). \tag{9}$$

The proof of Theorem 3.2 is deferred to Appendix A.

In practice, Eq. (8) is reformulated in its Lagrangian form, and a regularization parameter $\mu$ is used to control the strength of the sparsity-promoting constraint. Also, inspired by the $\beta$-VAE framework (Higgins *et al.*, 2017), an additional regularization parameter $\lambda$ was used to control the strength of the likelihood term $\log q_{\mathbf{n}}(\mathbf{g} - Hh_\phi(\boldsymbol{\zeta}_i; \mathbf{g}))$. This modifies the problem

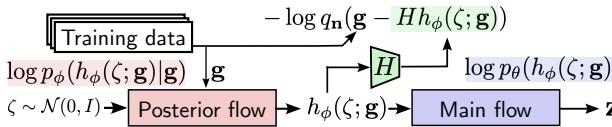

Figure 1: A schematic of the AmbientFlow framework

to maximizing the following objective function, which was optimized using gradient-based methods.

$$\tilde{\mathcal{L}}_M(\theta, \phi) = \mathbb{E}_{\mathbf{g} \sim q_{\mathbf{g}}} \mathbb{E}_{\boldsymbol{\zeta}_i \sim q_\zeta} \left[ \underset{0 < i \leq M}{\mathrm{logavgexp}} \left\{ \log p_\theta\big(h_\phi(\boldsymbol{\zeta}_i; \mathbf{g})\big) + \lambda \log q_{\mathbf{n}}\big(\mathbf{g} - Hh_\phi(\boldsymbol{\zeta}_i; \mathbf{g})\big) - \log p_\phi\big(h_\phi(\boldsymbol{\zeta}_i; \mathbf{g}) \,|\, \mathbf{g}\big) \right\} \right.$$
$$\left. - \mu \big\| \Phi h_\phi(\boldsymbol{\zeta}_i; \mathbf{g}) - \mathrm{proj}_{\mathcal{S}_k}(\Phi h_\phi(\boldsymbol{\zeta}_i; \mathbf{g})) \big\|_1 \right] \tag{10}$$

The proposed additive sparsifying penalty is computed by hard-thresholding the output of $\Phi h_\phi(\boldsymbol{\zeta}_i; \mathbf{g})$ to project it onto the space of $k$-sparse signals, and computing the $\ell_1$ norm of the residual, with $k$ and $\mu$ being treated as tunable hyperparameters. However, note that consistent with Eq. (10), the loss terms for both the INNs correspond to the original (un-thresholded) outputs of the posterior. This ensures that invertibility is maintained, and the loss terms from both the INNs are well-defined. Empirically, we observe that the proposed $\ell_1$ penalty also promotes sparse deviation of the output of $h_\phi$ from $\mathcal{S}_k$, which improves the quality of the images generated by AmbientFlow.

## 4 Numerical Studies

This section describes the numerical studies used to demonstrate the utility of AmbientFlow for learning object distributions from noisy and incomplete imaging measurements. The studies include toy problems in two dimensions, low-dimensional problems involving a distribution of handwritten digits from the MNIST dataset, problems involving face images from the CelebA-HQ dataset as well as the problem of recovering the object distribution from stylized magnetic resonance imaging measurements. A case study that demonstrates the utility of AmbientFlow in the downstream tasks of image reconstruction and posterior sampling is also

described. Additional numerical studies, including an evaluation of the posterior network, additional ablation studies, and a face image inpainting case study are included in Appendix B.

**Datasets and imaging models.**

*1) Toy problems:* First, a two-dimensional object distribution $q_{\mathbf{f}} : \mathbb{R}^2 \to \mathbb{R}$ was considered, which was created as a sum of eight Gaussian distributions $\mathcal{N}(\mathbf{c}_i, \sigma_{\mathbf{f}}^2 I_2)$, $1 \leq i \leq 8$, with centers $\mathbf{c}_i$ located at the vertices of a regular octagon centered at the origin, and standard deviation $\sigma_{\mathbf{f}} = 0.15$, as shown in Fig. 2a. The forward operator was the identity operator, and the noise $\mathbf{n}$ was distributed as a zero-mean Gaussian with covariance $\sigma_{\mathbf{n}}^2 I_2$, with $\sigma_{\mathbf{n}} = 0.45$. The distribution of the

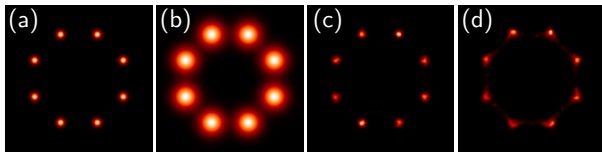

Figure 2: (a) True distribution $q_{\mathbf{f}}$, (b) distribution $q_{\mathbf{g}}$ of measurements, (c) distribution learned by a flow model trained on true objects, and (d) distribution learned by AmbientFlow trained on measurements.

measurements $\mathbf{g} = \mathbf{f} + \mathbf{n}$ is shown in Fig. 2b. A training dataset of size $D = 5 \times 10^7$ was used.

Next, a problem of recovering the distribution of MNIST digits from noisy and/or blurred images of MNIST digits was considered (LeCun *et al.*, 1998). For this problem, three different forward models were considered, namely the identity operator, and two Gaussian blurring operators $H_{\text{blur1}}$ and $H_{\text{blur2}}$ with root-mean-squared (RMS) width values $\sigma_b = 1.5$ and $3.0$ pixels. The measurement noise was distributed as $\mathbf{n} \sim \mathcal{N}(\mathbf{0}, \sigma_{\mathbf{n}}^2 I_m)$, with $\sigma_{\mathbf{n}} = 0.3$.

*2) Face image study:* For the face image study, images of size $n = 64 \times 64 \times 3$ from the CelebA-HQ dataset were considered (Karras *et al.*, 2017). Two forward models were considered, namely the identity operator and the blurring operator with RMS width $\sigma_b = 1.5$, and the measurement noise was distributed as $\mathbf{n} \sim \mathcal{N}(\mathbf{0}, \sigma_{\mathbf{n}}^2 I_m)$, $\sigma_{\mathbf{n}} = 0.2$. A discrete gradient operator was used as the sparsifying transform $\Phi$.

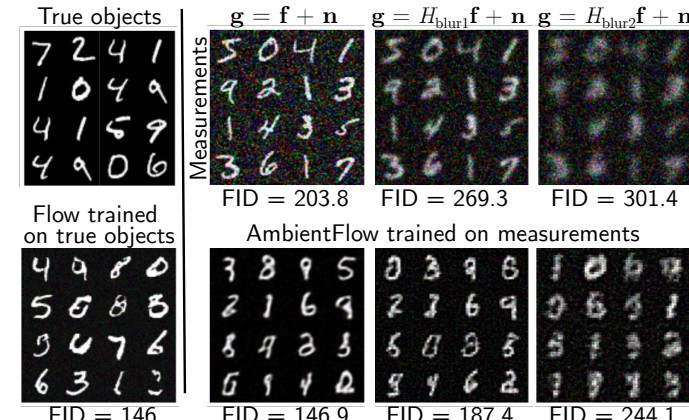

Figure 3: Samples from the flow model trained on true objects and AmbientFlow trained on measurements shown alongside samples of true objects and measurements for the MNIST dataset.

*3) Stylized MRI study:* In this study, the problem of recovering the distribution of objects from simulated, stylized MRI measurements was considered. T2-weighted brain images of size $n = 128 \times 128$ from the FastMRI initiative database were considered (Zbontar *et al.*, 2018) as samples from the object distribution. Stylized MRI measurements with undersampling ratio $n/m = 1$ (fully sampled) and $n/m = 4$ were simulated using the fast Fourier transform (FFT). Complex valued iid Gaussian measurement noise with standard deviation 0.1 times the total range of ground truth gray values was considered. A discrete gradient operator was used as the sparsifying transform $\Phi$.

**Network architecture and training.** [1] The architecture of the main flow model $G_\theta$ was adapted from the Glow architecture (Kingma & Dhariwal, 2018). The posterior network was adapted from the conditional INN architecture proposed by Ardizzone, *et. al* (Ardizzone *et al.*, 2021). AmbientFlow was trained using PyTorch using an NVIDIA Quadro RTX 8000 GPU. All hyperparameters for the main INN were fixed based on a PyTorch implementation of the Glow architecture (Seonghyeon), except the number of blocks, which was set to scale logarithmically by the image dimension.

---

[1]Our PyTorch implementation of AmbientFlow can be found at `https://github.com/comp-imaging-sci/ambientflow`

**Baselines and evaluation metrics.** For each dataset, an INN was trained on the ground-truth objects. The architecture and hyperparameters used for this INN for a particular dataset were identical to the ones used for main flow $G_\theta$ within the AmbientFlow framework trained on that dataset. Besides, for each forward model for the face image and stylized MRI study, a non-data-driven image restoration/reconstruction algorithm was used to generate a dataset of individual estimates of the object. For $H = I_m$ with $\mathbf{n} \sim \mathcal{N}(\mathbf{0}, \sigma_\mathbf{n}^2 I_m)$, the block-matching and 3D filtering (BM3D) denoising algorithm was used to perform image restoration (Dabov *et al.*, 2007). For the blurring operator with Gaussian noise, Wiener deconvolution was used for this purpose (Blahut, 2004). For the stylized MRI study, a penalized least-squares with TV regularization (PLS-TV) algorithm was used for image reconstruction (Beck & Teboulle, 2009). The regularization parameter for the image reconstruction method was tuned to give the lowest RMS error (RMSE) for every individual reconstructed image. Although this method of tuning the parameters is not feasible in real systems, it gives the best set of reconstructed images in terms of the RMSE, thus providing a strong baseline.

The Frechet Inception distance (FID) score, computed using the Clean-FID package (Parmar *et al.*, 2022), was used to compare a dataset of 5,000 true objects with an equivalent number of images synthesized using (1) the INN trained on the true objects and (2) the AmbientFlow trained on the measurements, and (3) images individually reconstructed from their corresponding measurements. Additionally, for the stylized MRI study, radiomic features meaningful to medical imaging were computed on the true objects, generated objects, and reconstructed images (Van Griethuysen *et al.*, 2017).

**Case study.** Next, the utility of the AmbientFlow in a downstream Bayesian inference task was examined. For this purpose, a case study of image reconstruction from incomplete measurements was considered, where the AmbientFlow was used as a prior. Importantly, we consider the scenario where the forward model used for simulating the measurements is different from the one associated with the AmbientFlow training. Preliminaries of generative priors for image reconstruction are discussed in (Dimakis *et al.*, 2022). In this study, the following two image reconstruction tasks are considered – (1) approximate maximum a posteriori (MAP) estimation, i.e. approximating the mode of the posterior $p_\theta(\cdot \,|\, \mathbf{g})$, and (2) approximate sampling from the posterior $p_\theta(\cdot \,|\, \mathbf{g})$.

For both the tasks, an AmbientFlow trained on the fully sampled, noisy simulated MRI measurements, as well as the flow trained on the true objects were considered. For the first task, the compressed sensing using generative models (CSGM) formalism was used to obtained approximate MAP estimates from measurements, for a held-out for a test dataset of size 45 (Asim *et al.*, 2020):

$$\hat{\mathbf{f}}_{\mathrm{MAP}} = G_\theta(\hat{\mathbf{z}}_{\mathrm{MAP}}), \quad \text{with } \hat{\mathbf{z}}_{\mathrm{MAP}} = \arg\min_{\mathbf{z}} \|\mathbf{g} - HG_\theta(\mathbf{z})\|_2^2 + \lambda \|\mathbf{z}\|_2^2. \tag{11}$$

For the second task, approximate posterior sampling was performed with the flow models as priors using annealed Langevin dynamics (ALD) iterations proposed in Jalal, *et al.* (Jalal *et al.*, 2021a). For each true object, the minimum mean-squared error (MMSE) estimate $\hat{\mathbf{f}}_{\mathrm{MMSE}}$ and the pixelwise standard deviation map $\hat{\sigma}$ were computed empirically using 40 samples obtained via ALD iterations. These image estimates were compared with reconstructed images obtained using the PLS-TV method. The regularization parameters for each image reconstruction method were tuned to achieve the best RMSE on a single validation image, and then kept constant for the entire test dataset.

## 5   Results

Figure 2 shows the true object distribution, the distribution learned by a flow model trained on objects, the measurement distribution, and the object distribution recovered by AmbientFlow trained using the measurements. It can be seen that AmbientFlow is successful in generating nearly noiseless samples that belong to one of the eight Gaussian blobs, although a small number of generated samples lie in the connecting region between the blobs.

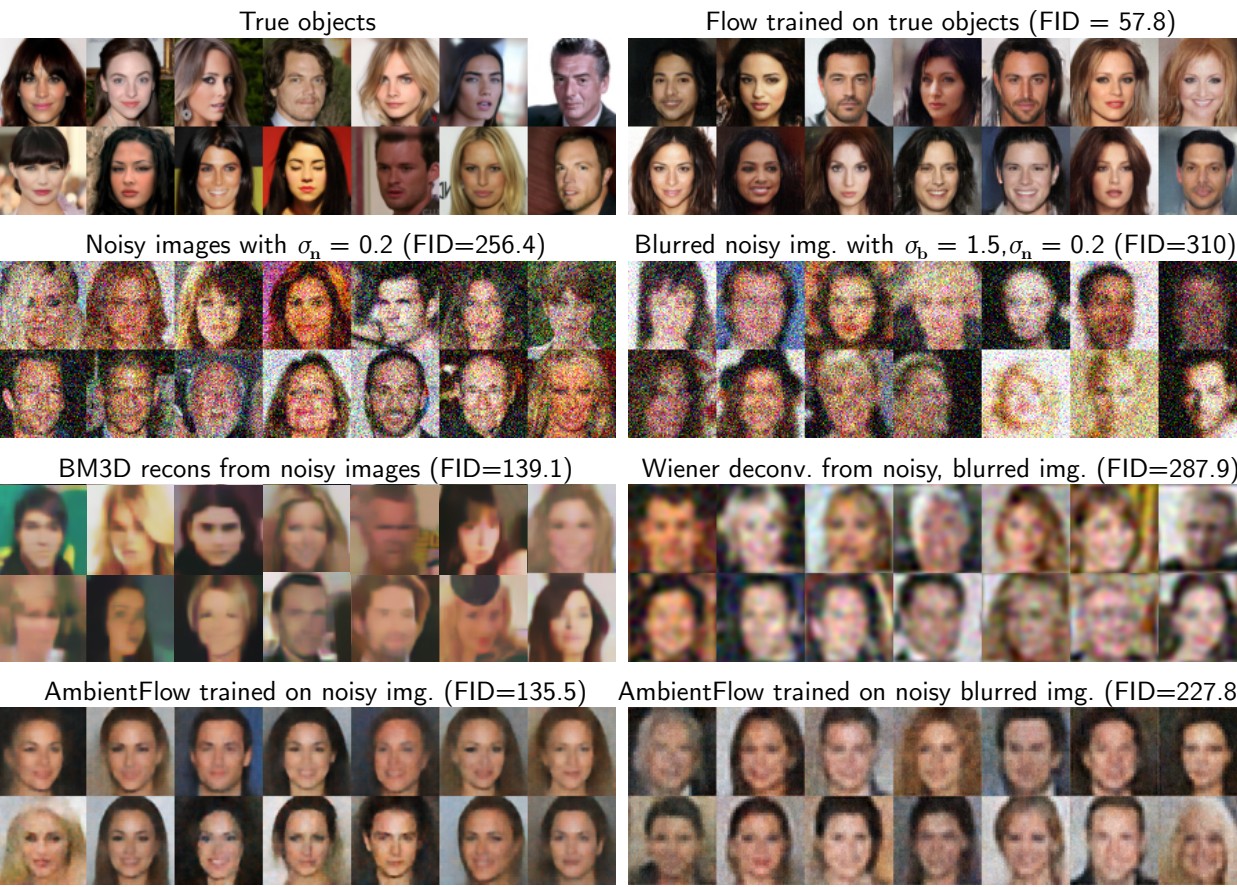

Figure 4: True objects, noisy/blurred image measurements, reconstructed images and images synthesized by the flow model trained on the true objects, as well as the AmbientFlows trained on the measurements for the CelebA-HQ dataset.

For the MNIST dataset, Fig. 3 shows samples from the flow model trained on true objects and AmbientFlow trained on measurements alongside samples of true objects and measurements, for different measurement models. It can be seen that when the degradation process is a simple noise addition, the AmbientFlow produces samples that visually appear denoised. When the degradation process consists of blurring and noise addition, the AmbientFlow produces images that do not contain blur or noise, although the visual image quality degrades as the blur increases. The visual findings are corroborated with quantitative results in terms of the FID score, shown in Fig. 3.

Figure 4 shows the results of the face image study for the two different measurement processes considered - (1) additive noise, and (2) Gaussian blur followed by additive noise. It can be seen that both visually and in terms of the FID score, the quality of images generated by the AmbientFlow models was second only to the flow trained directly on the true objects, for both the forward models considered. The images synthesized by the AmbientFlow models had better visual quality and better fidelity in distribution with the true objects with respect to FID than the ones produced by individually performing image restoration using BM3D and Wiener deconvolution for the two forward models respectively. This suggests that an AmbientFlow trained directly on the measurements would give a better approximation to the object distribution in terms of the considered metrics as compared to a regular flow model trained on the image datasets individually reconstructed via BM3D/Wiener deconvolution.

The results of the stylized MRI study are shown in Fig. 5. The visual and FID-based quality of images synthesized by the AmbientFlow models was inferior only to the images synthesized by the flow trained

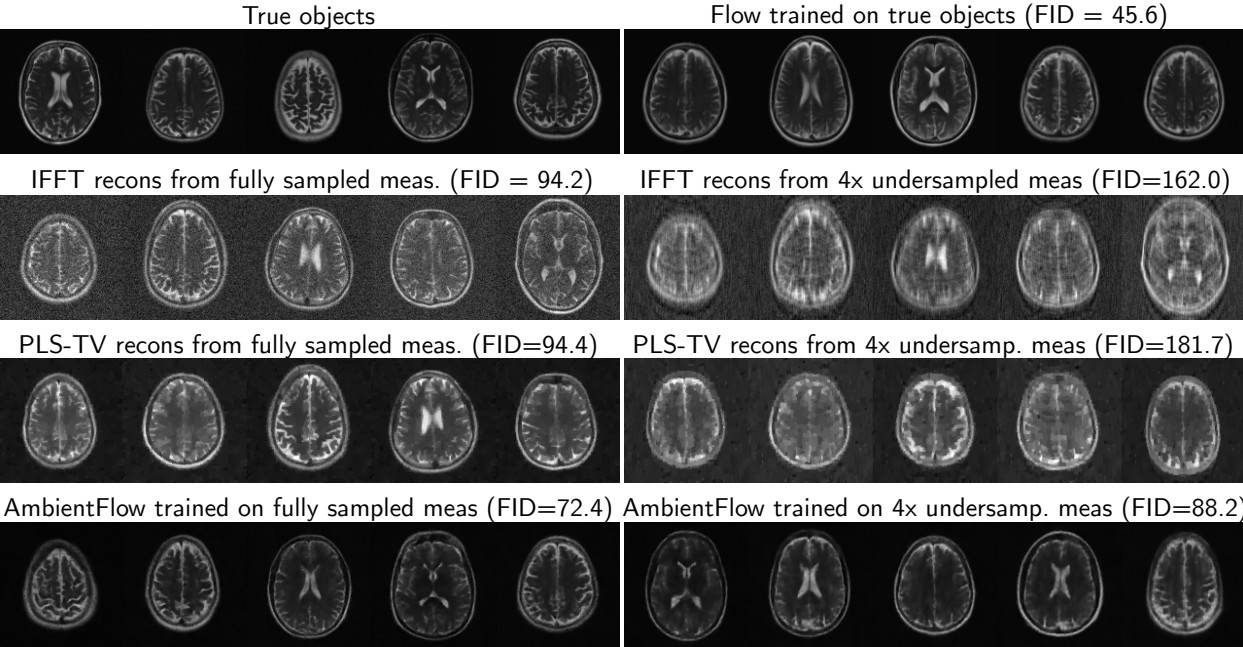

Figure 5: True objects, IFFT-based image estimates, PLS-TV based image estimates and images synthesized by the flow model trained on the true objects, as well as the AmbientFlows trained on the measurements for the stylized MRI study.

Table 1: The mean ± standard deviation of the RMSE and SSIM values computed over the following test image datasets – (1) the images reconstructed using the PLS-TV method, (2) the MAP and MMSE estimates using the flow prior trained on true objects, and (3) the MAP and MMSE estimates using the AmbientFlow prior trained on fully sampled noisy measurements.

| Method | | RMSE | SSIM |
|---|---|---|---|
| PLS-TV | | $0.038 \pm 0.007$ | $0.806 \pm 0.019$ |
| Flow prior trained on true objects | MAP Estimate | $0.025 \pm 0.005$ | $0.922 \pm 0.013$ |
| | MMSE Estimate | $\mathbf{0.022 \pm 0.003}$ | $\mathbf{0.940 \pm 0.006}$ |
| AmbientFlow prior | MAP Estimate | $0.025 \pm 0.004$ | $0.925 \pm 0.012$ |
| | MMSE Estimate | $\mathbf{0.022 \pm 0.004}$ | $\mathbf{0.936 \pm 0.008}$ |

directly on objects, and was superior to the images reconstructed individually from the measurements using the PLS-TV method. Since the underlying Inception network used to compute the FID score is not directly related to medical images, additional evaluation was performed in terms of radiomic features relevant to medical image assessments.

Figure 6 plots the empirical PDF over the first two principal components of the radiomic features extracted from each of the MR image sets shown in Fig. 5, except the IFFT image estimates. It can be seen that there is a significant disparity between the principal radiomic feature PDFs of the true objects and the images reconstructed individually using PLS-TV. On the other hand, the AmbientFlow-generated images have a radiomic feature distribution closer to the true objects for both the fully sampled and 4-fold undersampled cases. This implies that, training an AmbientFlow on the noisy/incomplete measurements yielded an estimate of the object distribution that was more accurate in terms of relevant radiomic features, than the one defined by images individually reconstructed from the measurements using the PLS-TV method.

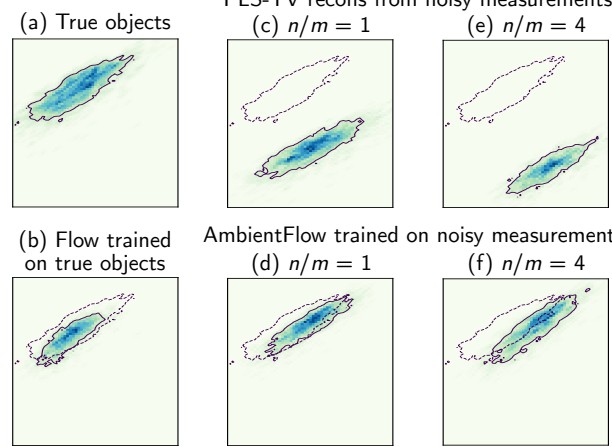

Figure 6: Empirical PDF over the first two principal components of the radiomic features extracted from MRI images. For each plot, the bold contour encloses the region containing 80% of the probability mass. For (b-f), the dotted contour encloses the region containing 80% of the probability mass of the true objects.

Next, the utility of the AmbientFlow for Bayesian inference is demonstrated with the help of an image reconstruction case study. Figure 7 shows a true object alongside images reconstructed from stylized 4-fold undersampled MR measurements simulated from the object, using the reconstruction methods described in Section 4. Recall that for both the flow-based priors shown, the MAP estimate was obtained using the CSGM framework (Bora *et al.*, 2017), and the MMSE estimate and the pixelwise standard deviation maps were computed empirically from samples from the posterior $p_\theta(\mathbf{f} \,|\, \mathbf{g})$ obtained using annealed Langevin dynamics (Jalal *et al.*, 2021a). Visually, it can be seen that the images reconstructed using the Ambient-Flow prior were comparable to the images reconstructed using the flow prior trained on the true objects, and better than the image reconstructed using the PLS-TV method. Table 1 shows the root-mean-squared error (RMSE) and structural similarity (SSIM) (Wang *et al.*, 2004) of the reconstructed images with respect to the true object, averaged over a dataset of 45 test images. It can be seen that in terms of RMSE and SSIM, both the MAP and the MMSE image estimates obtained using the AmbientFlow prior are comparable to those obtained using the flow prior trained on true objects, despite the AmbientFlow being trained only using noisy stylized MR measurements.

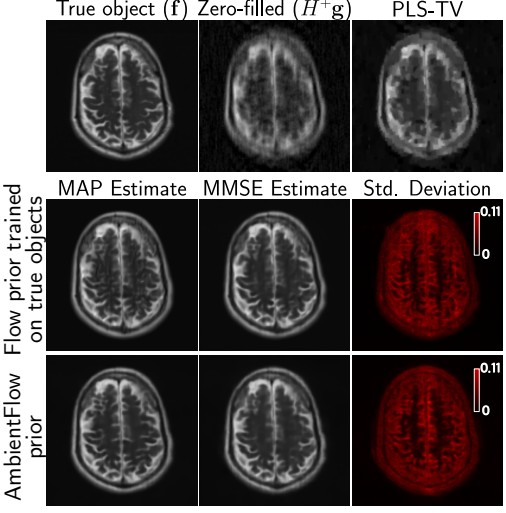

Figure 7: Image estimates and pixelwise standard deviation maps from the image reconstruction case study.

## 6 Discussion and conclusion

An estimate of the distribution of objects is known to be important for applications in imaging science. This is because an unconditional generative model of objects can potentially be used in image reconstruction tasks without the need for paired data of images and measurements, and in a way that accommodates a

wide variety of relevant forward models (Bora *et al.*, 2017; Asim *et al.*, 2020). Additionally, unconditional generative models can be used to approximate ideal Bayesian classifiers (Zhou *et al.*, 2023), or for anomaly detection (Zhao *et al.*), or image manipulation (Torres & Brefeld). However, obtaining such an estimate in terms of a generative model that is useful for downstream tasks has remained challenging, especially when only noisy and incomplete measurements of the objects are available. In this work, a framework for learning flow-based generative models of objects from noisy/incomplete measurements was developed. The presented numerical studies show that the proposed AmbientFlow framework was able to mitigate the effects of data incompleteness and noise present in the measurements in order to build an accurate estimate of the object distribution in terms of the considered evaluation metrics, and generate visually appealing images. In terms of perceptual measures such as the FID score as well as domain specific radiomic features, the images synthesized by AmbientFlow maintained higher distributional fidelity with the true objects than the images individually reconstructed from the measurements. Furthermore, when the AmbientFlow trained on noisy measurements was employed as a prior in an image reconstruction task, the image estimates obtained were as accurate in terms of RMSE and SSIM as those obtained when a flow model trained directly on the true objects was employed.

Some of the above observations also apply to the AmbientGAN framework developed by Bora, *et al.* (Bora *et al.*, 2018). Although the current state-of-the-art GANs may lead to images with better perceptual quality than IGMs, GANs trained on medical images have been shown to misrepresent medically relevant statistics despite producing visually appealing images (Kelkar *et al.*, 2023a). Also, an important drawback of GANs with in the context of imaging science is its unreliability in downstream Bayesian inference tasks. For example, GAN-constrained image reconstruction methods are known to be prone to hallucinations caused by dataset bias, distribution shifts and representation error, whereas IGMs used for the same purpose have been shown to be comparatively robust (Bhadra *et al.*, 2021; Asim *et al.*, 2020; Menon *et al.*, 2020; Kelkar *et al.*, 2021; Jalal *et al.*, 2021b;a; Zhao *et al.*, 2022). These drawbacks of GANs are in part due to the insufficient representation capacity and inability to access the log-probability, both of which are applicable for GANs learned in the ambient setting as well. IGMs, on the other hand, due to their ability to accurately represent images and compute fast, exact density estimates, are more suitable for some downstream inference tasks such as image reconstruction, posterior sampling and uncertainty quantification as compared to GANs. Further similarities and differences between AmbientGAN and AmbientFlow are as follows. In some scenarios, AmbientGAN can be more flexible since it can easily accommodate arbitrary differentiable random forward models, whereas AmbientFlow is limited by the capabilities of the posterior network. However, in the presence of a fixed null space, some studies have shown that AmbientGAN performance can degrade and the images produced by it can display aliasing artifacts characteristic of the measurement operator (Zhou *et al.*, 2022). To the best of our knowledge, incorporating additional priors into the AmbientGAN loss function has not been rigorously studied. Another key difference between the two approaches is that unlike AmbientGAN, AmbientFlow also provides a posterior network which can be useful by itself for certain image reconstruction tasks, as shown in Appendix B.

The AmbientFlow framework bears some similarity with variational autoencoders (VAEs). In both cases, a variational lower bound of the evidence of the data is minimized. The object distribution in AmbientFlow is analogous to the latent variable distribution in VAEs. However, in VAEs, the latent distribution is typically simple and non-unique, and its desirable properties include disentanglement, tractable sampling, and accurate representation of the data via a trained decoder. However, in the AmbientFlow framework, the object distribution is a complex high dimensional distribution that is of primary interest and needs to be recovered as accurately as possible. It is related to the measured data via a physical measurement process, and may have known structure such as transform compressibility. Therefore, the aims and objectives of the two frameworks that guide their design are radically different.

The posterior network in our work also has interesting connections to deep probabilistic imaging (DPI) (Sun & Bouman, 2021), which also uses an invertible network to model the posterior, and trains it in a way that is constrained by a prior. However, unlike DPI, the posterior model in our work is an end-to-end conditional generative model that, when trained, can directly produce posterior samples by taking in the measurement vector as one of the inputs.

The presented framework can be adopted to other generative models that utilize a log-likelihood-based training objective, such as denoising diffusion probabilistic models (DDPMs) which enable high-quality Bayesian inference in imaging (Song *et al.*, 2021). Recent examples of learning diffusion models from noisy/incomplete data include (Aali *et al.*, 2023) and (Daras *et al.*, 2023). However, the approach by Aali *et al.* applies only to measurements with white Gaussian noise and an identity forward operator. In contrast, the approach by Daras, *et al.* can incorporate a wide class of forward operators, but does not account for measurement noise. In the future, an extension of the AmbientFlow framework to diffusion models could account for both, forward operators with a null space, as well as measurement noise.

A key limitation of the proposed framework is that its performance depends heavily on the design of the posterior network. The posterior network architecture currently employed is inspired by Ardizzone, *et. al* (Ardizzone *et al.*, 2021). It displays favourable inductive biases for images due to several design choices such as wavelet-based downsampling layers, and a Laplacian pyramid feature extractor for the conditioning input. However, this architecture may not be able to properly model the posterior when $H$ depends on a random parameter, such as a random view angle relevant for cryo-electron microscopy (Zhong *et al.*, 2021). In theory, it is straightforward to modify the loss function so that in addition to $\mathbf{g}$, a random forward model $H \sim q_H$ is also a conditioning input to the posterior network. However, in practice, designing a posterior network architecture that can successfully account for the random forward operator is nontrivial, and could be an interesting avenue for future work. Also, although this work involves preliminary assessments of AmbientFlow using the FID score and radiomic features, a proper evaluation of generative models for imaging applications is still an open problem (Sankaranarayanan *et al.*, 2023). Thorough evaluation of such models would involve assessing whether they can reproduce image statistics that are relevant to a wide variety of downstream tasks (Kelkar *et al.*, 2023a;b).

## Acknowledgements

This work was supported by the National Institute of Health (NIH) award EB034249. We acknowledge Carl Edwards for useful discussions.

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

## A   Theoretical analysis for Section 3

First, the notation defined in Section 3 is rehashed here for convenience.

**Notation.**   Let $q_{\mathbf{f}}$, $q_{\mathbf{g}}$ and $q_{\mathbf{n}}$ denote the unknown true object distribution to-be-recovered, the true measurement distribution and the known measurement noise distribution, respectively. Let $\mathcal{D} = \{\mathbf{g}^{(i)}\}_{i=1}^{D}$ be a dataset of independent and identically distributed (iid) measurements drawn from $q_{\mathbf{g}}$. Let $G_\theta : \mathbb{R}^n \to \mathbb{R}^n$ be an INN. Let $p_\theta$ be the distribution represented by $G_\theta$, i.e. given a latent distribution $q_{\mathbf{z}} = \mathcal{N}(\mathbf{0}, I_n)$, $G_\theta(\mathbf{z}) \sim p_\theta$ for $\mathbf{z} \sim q_{\mathbf{z}}$. Also, let $\psi_\theta$ be the distribution of fake measurements, i.e. for $\mathbf{f} \sim p_\theta$, $H\mathbf{f} + \mathbf{n} \sim \psi_\theta$. Let $p_\theta(\mathbf{f} \,|\, \mathbf{g}) \propto q_{\mathbf{n}}(\mathbf{g} - H\mathbf{f}) \, p_\theta(\mathbf{f})$ denote the posterior induced by the learned object distribution represented by $G_\theta$. Let $\Phi \in \mathbb{R}^{l \times n}$, $l \geq n$ be a full-rank linear transformation (henceforth referred to as a sparsifying transform). Also, let $\mathcal{S}_k = \{\mathbf{v} \in \mathbb{R}^n \text{ s.t. } \|\Phi \mathbf{v}\|_0 \leq k\}$ be the set of vectors $k$-sparse with respect to $\Phi$. Since $\Phi$ is full-rank, throughout this chapter we assume without the loss of generality, that $\|\Phi^+\|_2 \leq 1$, where $\Phi^+$ is the Moore-Penrose pseudoinverse of $\Phi$. Throughout the manuscript, we also assume that $q_{\mathbf{f}}$ is absolutely continuous with respect to $p_\theta$, and $q_{\mathbf{g}}$ is absolutely continuous with respect to $\psi_\theta$.

### A.1   Proof of Theorem 3.1

**Theorem 3.1.**   *Let $h_\phi$ be such that $p_\phi(\mathbf{f} \,|\, \mathbf{g}) > 0$ over $\mathbb{R}^n$. Minimizing $D_{\mathrm{KL}}(q_{\mathbf{g}}\|\psi_\theta)$ is equivalent to maximizing the objective function $\mathcal{L}(\theta, \phi)$ over $\theta, \phi$, where $\mathcal{L}(\theta, \phi)$ is defined as*

$$\mathcal{L}(\theta, \phi) = \mathbb{E}_{\mathbf{g} \sim q_{\mathbf{g}}} \left[ \log \mathbb{E}_{\boldsymbol{\zeta} \sim q_{\zeta}} \left\{ \frac{p_\theta\big(h_\phi(\boldsymbol{\zeta}; \mathbf{g})\big) \, q_{\mathbf{n}}\big(\mathbf{g} - H h_\phi(\boldsymbol{\zeta}; \mathbf{g})\big)}{p_\phi\big(h_\phi(\boldsymbol{\zeta}; \mathbf{g}) \,|\, \mathbf{g}\big)} \right\} \right] \tag{12}$$

*Proof.* From the definition of KL divergence, we have

$$D_{\mathrm{KL}}(q_{\mathbf{g}}\|\psi_\theta) = \mathbb{E}_{\mathbf{g} \sim q_{\mathbf{g}}} \left[ \log \frac{q_{\mathbf{g}}(\mathbf{g})}{\psi_\theta(\mathbf{g})} \right] \tag{13}$$

$$= \mathbb{E}_{\mathbf{g} \sim q_{\mathbf{g}}} \log q_{\mathbf{g}}(\mathbf{g}) - \mathbb{E}_{\mathbf{g} \sim q_{\mathbf{g}}} \log \psi_\theta(\mathbf{g}). \tag{14}$$

Now, $\psi_\theta(\mathbf{g})$ can be written as

$$\psi_\theta(\mathbf{g}) = \int q_{\mathbf{g}|\mathbf{f}}(\mathbf{g}|\mathbf{f}) p_\theta(\mathbf{f}) d\mathbf{f} \tag{15}$$

$$= \int p_\phi(\mathbf{f}|\mathbf{g}) \frac{q_{\mathbf{n}}(\mathbf{g} - H\mathbf{f}) \, p_\theta(\mathbf{f})}{p_\phi(\mathbf{f}|\mathbf{g})} d\mathbf{f} \tag{16}$$

$$= \mathbb{E}_{\mathbf{f} \sim p_\phi(\cdot|\mathbf{g})} \left[ \frac{q_{\mathbf{n}}(\mathbf{g} - H\mathbf{f}) \, p_\theta(\mathbf{f})}{p_\phi(\mathbf{f}|\mathbf{g})} \right]. \tag{17}$$

Therefore,

$$\mathbb{E}_{\mathbf{g} \sim q_{\mathbf{g}}} \log \psi_\theta(\mathbf{g}) = \mathbb{E}_{\mathbf{g} \sim q_{\mathbf{g}}} \left[ \log \mathbb{E}_{\mathbf{f} \sim p_\phi(\cdot|\mathbf{g})} \left\{ \frac{p_\theta(\mathbf{f}) \, q_{\mathbf{n}}(\mathbf{g} - H\mathbf{f})}{p_\phi(\mathbf{f} \,|\, \mathbf{g})} \right\} \right] \tag{18}$$

$$= \mathbb{E}_{\mathbf{g} \sim q_{\mathbf{g}}} \left[ \log \mathbb{E}_{\boldsymbol{\zeta} \sim q_{\zeta}} \left\{ \frac{p_\theta\big(h_\phi(\boldsymbol{\zeta}; \mathbf{g})\big) \, q_{\mathbf{n}}\big(\mathbf{g} - H h_\phi(\boldsymbol{\zeta}; \mathbf{g})\big)}{p_\phi\big(h_\phi(\boldsymbol{\zeta}; \mathbf{g}) \,|\, \mathbf{g}\big)} \right\} \right] \tag{19}$$

$$= \mathcal{L}(\theta, \phi). \tag{20}$$

Therefore,

$$D_{\mathrm{KL}}(q_{\mathbf{g}}\|\psi_\theta) = \mathbb{E}_{\mathbf{g} \sim q_{\mathbf{g}}} \log q_{\mathbf{g}}(\mathbf{g}) - \mathcal{L}(\theta, \phi). \tag{21}$$

Since $\mathbb{E}_{\mathbf{g} \sim q_{\mathbf{g}}} \log q_{\mathbf{g}}(\mathbf{g})$ is a constant, minimizing $D_{\mathrm{KL}}(q_{\mathbf{g}}\|\psi_\theta)$ is equivalent to maximizing $\mathcal{L}(\theta, \phi)$. $\qquad \square$

**Remark.** As described in Section 3, a variational lower bound of $\mathcal{L}$ is optimized in this work:

$$\mathcal{L}_M(\theta, \phi) = \mathbb{E}_{\mathbf{g}, \boldsymbol{\zeta}_i} \underset{0 < i \leq M}{\text{logavgexp}} \left[ \log p_\theta \big( h_\phi(\boldsymbol{\zeta}_i; \mathbf{g}) \big) + \log q_\mathbf{n} \big( \mathbf{g} - H h_\phi(\boldsymbol{\zeta}_i; \mathbf{g}) \big) - \log p_\phi \big( h_\phi(\boldsymbol{\zeta}_i; \mathbf{g}) \mid \mathbf{g} \big) \right], \quad (22)$$

where $\boldsymbol{\zeta}_i \sim q_{\boldsymbol{\zeta}}$, $0 < i \leq M$, and $\text{logavgexp}_{0 < i \leq M}(x_i) := \log \left[ \frac{1}{M} \sum_{i=1}^M \exp(x_i) \right]$.

For sufficiently expressive parametrizations for $p_\theta$ and $h_\phi$, the maximum possible value of $\mathcal{L}_M$ is $\mathbb{E}_{\mathbf{g} \sim q_\mathbf{g}} \log q_\mathbf{g}(\mathbf{g})$, which corresponds to the scenario where the learned posteriors are consistent, i.e. $p_\phi(\mathbf{f} \mid \mathbf{g}) = p_\theta(\mathbf{f} \mid \mathbf{g})$, and the learned distribution of measurements matches the true measurement distribution, i.e. $\psi_\theta = q_\mathbf{g}$. This can be shown as follows.

For $M = 1$, $\mathcal{L}_M(\theta, \phi)$ reduces to the evidence lower bound (ELBO):

$$\mathcal{L}_{\text{ELBO}}(\theta, \phi) = \mathbb{E}_{\mathbf{g} \sim q_\mathbf{g}, \boldsymbol{\zeta} \sim q_{\boldsymbol{\zeta}}} \left[ \log p_\theta \big( h_\phi(\boldsymbol{\zeta}; \mathbf{g}) \big) + \log q_\mathbf{n} \big( \mathbf{g} - H h_\phi(\boldsymbol{\zeta}; \mathbf{g}) \big) \right.$$
$$\left. - \log p_\phi \big( h_\phi(\boldsymbol{\zeta}; \mathbf{g}) \mid \mathbf{g} \big) \right]. \quad (23)$$
$$= \mathbb{E}_{\mathbf{g} \sim q_\mathbf{g}} \log \psi_\theta(\mathbf{g}) - D_{\text{KL}}(p_\phi(\cdot|\mathbf{g}) \| p_\theta(\cdot|\mathbf{g})). \quad (24)$$

Similar to importance-weighted autoencoders (IWAE), (Burda *et al.*, 2015), it can be shown that

$$\mathcal{L}_{\text{ELBO}}(\theta, \phi) \leq \mathcal{L}_M(\theta, \phi) \leq \mathbb{E}_{\mathbf{g} \sim q_\mathbf{g}} \log \psi_\theta(\mathbf{g}) \leq \mathbb{E}_{\mathbf{g} \sim q_\mathbf{g}} \log q_\mathbf{g}(\mathbf{g}), \quad (25)$$

with equality occurring when $\psi_\theta = q_\mathbf{g}$ and $p_\phi(\cdot|\mathbf{g}) = p_\theta(\cdot|\mathbf{g})$. Thus the maximum value that can be achieved by $\mathcal{L}_M(\theta, \phi)$ is $\mathbb{E}_{\mathbf{g} \sim q_\mathbf{g}} \log q_\mathbf{g}(\mathbf{g})$.

**Lemma 3.1.** *If $H$ is a square matrix ($n = m$) with full-rank, if the noise $\mathbf{n}$ is independent of the object, and if the characteristic function of the noise $\chi_\mathbf{n}(\mathbf{v}) = \mathbb{E}_{\mathbf{n} \sim q_\mathbf{n}} \exp(\iota \mathbf{v}^\top \mathbf{n})$ has full support over $\mathbb{R}^m$ ($\iota$ is the square-root of $-1$), then $\psi_\theta = q_\mathbf{g} \Rightarrow p_\theta = q_\mathbf{f}$.*

*Proof.* This proof as been adapted from the AmbientGAN work (Bora *et al.*, 2018). Let $\mathbf{y} = H\mathbf{f}$ represent the noiseless measurements. Therefore,

$$\mathbf{g} = \mathbf{y} + \mathbf{n}, \quad (26)$$
$$\Rightarrow q_\mathbf{g} = q_\mathbf{y} * q_\mathbf{n}, \quad (27)$$

where $*$ represents a convolution (in the sense of linear systems theory) (Lathi & Green, 2005). Therefore,

$$\chi_\mathbf{g}(\mathbf{v}) = \chi_\mathbf{y}(\mathbf{v}) \chi_\mathbf{n}(\mathbf{v}), \quad \mathbf{v} \in \mathbb{R}^m. \quad (28)$$

Since $\chi_\mathbf{n}$ has full support over $\mathbb{R}^m$, $\chi_\mathbf{g}$ uniquely determines $\chi_\mathbf{y}$. Therefore, $q_\mathbf{g}$ uniquely determines $q_\mathbf{y}$.

Also, since $H$ is bijective, $q_\mathbf{y}$ uniquely determines $q_\mathbf{f}$. Therefore, $\psi_\theta = q_\mathbf{g} \Rightarrow p_\theta = q_\mathbf{f}$. $\qquad \square$

## A.2  Proof of Theorem 3.2

In order to prove Theorem 3.2, we first establish essential notation and intermediate results needed. Specifically, in Lemma A.1, we derive an expression for the wasserstein distance between a distribution of a random variable, and the distribution of its projection onto a set. We then proceed to prove Theorem 3.2.

**Notation.** For a closed set $\mathcal{S} \subset \mathbb{R}^n$, let $\text{proj}_\mathcal{S}(\mathbf{f})$ denote the orthogonal projection of $\mathbf{f}$ onto $\mathcal{S}$, defined as

$$\text{proj}_\mathcal{S}(\mathbf{f}) = \min_{\mathbf{f}' \in \mathcal{S}} \|\mathbf{f}' - \mathbf{f}\|_2 \quad (29)$$

For a PDF $q : \mathbb{R}^n \to \mathbb{R}$, let $q^\mathcal{S}$ denote the distribution of $\text{proj}_S(\mathbf{x})$, for $\mathbf{x} \sim q$. Also, for distributions $q_1, q_2$, let

$$W_1(q_1 \| q_2) := \inf_{\gamma \in \Gamma(q_1, q_2)} \mathbb{E}_{(\mathbf{x}_1, \mathbf{x}_2) \sim \gamma} \|\mathbf{x}_1 - \mathbf{x}_2\|_2 \quad (30)$$

denote the Wasserstein 1-distance, with $\Gamma(q_1, q_2)$ being the set of all joint distributions $\gamma : \mathbb{R}^{n \times n} \to \mathbb{R}$ with marginals $q_1, q_2$, i.e.

$$\int \gamma(\mathbf{x_1}, \mathbf{x_2}) d\mathbf{x_2} = q_1(\mathbf{x_1}), \quad \int \gamma(\mathbf{x_1}, \mathbf{x_2}) d\mathbf{x_1} = q_2(\mathbf{x_2}). \tag{31}$$

**Lemma A.1.** *Let $\mathbf{x} \in \mathbb{R}^n$ be a random vector with distribution $q$. Then, with the above notation,*

$$W_1(q \| q^{\mathcal{S}}) = \mathbb{E}_{\mathbf{x} \sim q} \| \mathbf{x} - \mathrm{proj}_{\mathcal{S}}(\mathbf{x}) \|_2. \tag{32}$$

*Proof.* Let $\gamma_0 : \mathbb{R}^{n \times n} \to \mathbb{R}$ be a degenerate joint distribution given by

$$\gamma_0(\mathbf{x}, \mathbf{w}) = q(\mathbf{x}) \delta (\mathbf{w} - \mathrm{proj}_{\mathcal{S}}(\mathbf{x})), \quad \mathbf{x}, \mathbf{w} \in \mathbb{R}^n, \tag{33}$$

where, $\delta(\mathbf{w})$ denotes the Dirac delta. Therefore, by definition of the Wasserstein distance,

$$W_1(q \| q^{\mathcal{S}}) \le \mathbb{E}_{(\mathbf{x}, \mathbf{w}) \sim \gamma_0} \| \mathbf{x} - \mathbf{w} \|_2, \tag{34}$$

$$= \int q(\mathbf{x}) \delta (\mathbf{w} - \mathrm{proj}_{\mathcal{S}}(\mathbf{x})) \| \mathbf{x} - \mathbf{w} \|_2 d\mathbf{x} d\mathbf{w}, \tag{35}$$

$$= \int q(\mathbf{x}) \| \mathbf{x} - \mathrm{proj}_{\mathcal{S}}(\mathbf{x}) \|_2 d\mathbf{x}, \tag{36}$$

$$= \mathbb{E}_{\mathbf{x} \sim q} \| \mathbf{x} - \mathrm{proj}_{\mathcal{S}}(\mathbf{x}) \|_2. \tag{37}$$

On the other hand, by definition of orthogonal projection,

$$\| \mathbf{x} - \mathrm{proj}_{\mathcal{S}}(\mathbf{x}) \|_2 \le \| \mathbf{x} - \mathbf{w} \|_2, \quad \forall \, \mathbf{w} \in \mathrm{supp}(q^{\mathcal{S}}). \tag{38}$$

Therefore,

$$\mathbb{E}_{\mathbf{x} \sim q} \| \mathbf{x} - \mathrm{proj}_{\mathcal{S}}(\mathbf{x}) \|_2 \le \mathbb{E}_{(\mathbf{x}, \mathbf{w}) \sim \gamma} \| \mathbf{x} - \mathbf{w} \|_2, \quad \gamma \in \Gamma(q, q^{\mathcal{S}}). \tag{39}$$

$$\Rightarrow \mathbb{E}_{\mathbf{x} \sim q} \| \mathbf{x} - \mathrm{proj}_{\mathcal{S}}(\mathbf{x}) \|_2 \le \inf_{\gamma \in \Gamma(q, q^{\mathcal{S}})} \mathbb{E}_{(\mathbf{x}, \mathbf{w}) \sim \gamma} \| \mathbf{x} - \mathbf{w} \|_2, \tag{40}$$

$$= W_1(q \| q^{\mathcal{S}}). \tag{41}$$

Equations (37) and (41) imply

$$W_1(q \| q^{\mathcal{S}}) = \mathbb{E}_{\mathbf{x} \sim q} \| x - \mathrm{proj}_{\mathcal{S}}(\mathbf{x}) \|_2. \tag{42}$$

$\square$

With all the tools in place, we now proceed to prove Theorem 3.2. Consider the optimization problem stated in Eq. (8):

$$\hat{\theta}, \hat{\phi} = \arg \min_{\theta, \phi} -\mathcal{L}_M(\theta, \phi) \quad \text{subject to} \quad \mathbb{E}_{\mathbf{g} \sim q_{\mathbf{g}}} \mathbb{E}_{\mathbf{f} \sim p_\phi(\cdot | \mathbf{g})} \| \Phi \mathbf{f} - \Phi \mathrm{proj}_{\mathcal{S}_k}(\mathbf{f}) \|_1 < \epsilon. \tag{8}$$

**Theorem 3.2.** *If the following hold:*
1. *$W_1(q_{\mathbf{f}} \| q_{\mathbf{f}}^{\mathcal{S}_k}) \le \epsilon'$ (the true object distribution is concentrated on $k$-sparse objects under $\Phi$),*
2. *$H$ satisfies the RIP for objects $k$-sparse w.r.t. $\Phi$, with isometry constant $\delta_k$,*
3. *the characteristic function of noise $\chi_{\mathbf{n}}(\mathbf{v})$ has full support over $\mathbb{C}^m$, and*
4. *$(\theta, \phi)$ satisfying $p_\theta = q_{\mathbf{f}}$ and $p_\phi(\cdot | \mathbf{g}) = p_\theta(\cdot | \mathbf{g})$ is a feasible solution to Eq. (8) ($G_\theta$ and $h_\phi$ have sufficient capacity),*

*then the distrubution $p_{\hat{\theta}}$ recovered via Eq. (8) is close to the true object distribution, in terms of the Wasserstein distance i.e.*

$$W_1(p_{\hat{\theta}} \| q_{\mathbf{f}}) \le \left( 1 + \frac{1}{\sqrt{1 - \delta_k}} \| H \|_2 \right) (\epsilon + \epsilon'). \tag{43}$$

The intuitive idea behind the proof of this theorem is as follows. Compressed sensing stipulates that under precribed conditions, the forward operator is injective on a set of sparse vectors. Thus, if an object distribution is sparse, then the distribution of its measurements should be uniquely linked to it. If the object distribution $q_{\mathbf{f}}$ is compressible, and if it is ensured that a compressible distribution $p_{\hat{\theta}}$ is recovered via Eq. (8), then both $q_{\mathbf{f}}$ and $p_{\hat{\theta}}$ will be concentrated on the sparse vectors, and will associated with the same measurement distribution $q_{\mathbf{g}}$. Since the sparse vectors are uniquely determined by the measurements, $p_{\hat{\theta}}$ and $q_{\mathbf{f}}$ must themselves be close.

*Proof.* Since $(\theta, \phi)$ satisfying $p_\theta = q_{\mathbf{f}}$, and $p_\phi(\cdot|\mathbf{g}) = p_\theta(\cdot|\mathbf{g})$ is a feasible solution to Eq. (8), the maximum value of $\mathcal{L}_M$ under Eq. (8) is $\mathbb{E}_{\mathbf{g} \sim q_{\mathbf{g}}} \log q_{\mathbf{g}}(\mathbf{g})$. Therefore, according to Eq. (25), for the estimated $\hat{\theta}$ and $\hat{\phi}$, $\mathcal{L}(\hat{\theta}, \hat{\phi}) = \mathbb{E}_{\mathbf{g} \sim q_{\mathbf{g}}} \log q_{\mathbf{g}}(\mathbf{g})$,

$$\psi_{\hat{\theta}} = q_{\mathbf{g}} \text{ and } p_{\hat{\phi}}(\cdot|\mathbf{g}) = p_{\hat{\theta}}(\cdot|\mathbf{g}). \tag{44}$$

Let $\mathbf{f}_1, \mathbf{f}_2 \in \mathbb{R}^n$. Therefore, by triangle inequality,

$$\|\mathbf{f}_1 - \mathbf{f}_2\|_2 = \left\|\mathbf{f}_1 - \mathbf{f}_1^{\mathcal{S}} + \mathbf{f}_2^{\mathcal{S}} - \mathbf{f}_2 + \mathbf{f}_1^{\mathcal{S}} - \mathbf{f}_2^{\mathcal{S}}\right\|_2, \tag{45}$$

$$\leq \left\|\mathbf{f}_1 - \mathbf{f}_1^{\mathcal{S}}\right\|_2 + \left\|\mathbf{f}_2^{\mathcal{S}} - \mathbf{f}_2\right\|_2 + \left\|\mathbf{f}_1^{\mathcal{S}} - \mathbf{f}_2^{\mathcal{S}}\right\|_2, \tag{46}$$

where $\mathbf{f}^{\mathcal{S}}$ is a shorthand for $\text{proj}_{\mathcal{S}_k}(\mathbf{f})$ for $\mathbf{f} \in \mathbb{R}^n$.

$\mathbf{f}_1, \mathbf{f}_2$ can be represented in terms of the spasifying transform $\Phi$. Let $\mathbf{c}_i = \Phi \mathbf{f}_i$ and $\mathbf{c}_i^{\mathcal{S}} = \Phi \mathbf{f}_i^{\mathcal{S}}$, for $i = 1, 2$. Therefore,

$$\|\mathbf{f}_1 - \mathbf{f}_2\|_2 \leq \left\|\mathbf{f}_1 - \mathbf{f}_1^{\mathcal{S}}\right\|_2 + \left\|\Phi^+\right\|_2 \left\|\mathbf{c}_2 - \mathbf{c}_2^{\mathcal{S}}\right\|_2 + \left\|\mathbf{f}_1^{\mathcal{S}} - \mathbf{f}_2^{\mathcal{S}}\right\|_2, \tag{47}$$

where $\Phi^+$ is the Moore-Penrose pseudoinverse of $\Phi$. Also, by definition, recall that $\mathbf{c}_1^{\mathcal{S}}$ and $\mathbf{c}_2^{\mathcal{S}}$ have at most $k$ non-zero values.

Now, let $\mathbf{y}_i = H\mathbf{f}_i$ for $i = 1, 2$. Therefore,

$$\left\|\mathbf{y}_1^{\mathcal{S}} - \mathbf{y}_2^{\mathcal{S}}\right\|_2 = \left\|\mathbf{y}_1^{\mathcal{S}} - \mathbf{y}_1 + \mathbf{y}_2 - \mathbf{y}_2^{\mathcal{S}} + \mathbf{y}_1 - \mathbf{y}_2\right\|_2 \tag{48}$$

$$\leq \left\|\mathbf{y}_1 - \mathbf{y}_1^{\mathcal{S}}\right\|_2 + \left\|\mathbf{y}_2 - \mathbf{y}_2^{\mathcal{S}}\right\|_2 + \left\|\mathbf{y}_1 - \mathbf{y}_2\right\|_2, \tag{49}$$

$$\leq \|H\|_2 \left\|\mathbf{f}_1 - \mathbf{f}_1^{\mathcal{S}}\right\|_2 + \left\|H\Phi^+\right\|_2 \left\|\mathbf{c}_2 - \mathbf{c}_2^{\mathcal{S}}\right\|_2 + \left\|\mathbf{y}_1 - \mathbf{y}_2\right\|_2. \tag{50}$$

Now, the restricted isometry property (RIP) on $H$ defined in Definition 2.1 implies

$$\left\|\mathbf{f}_1^{\mathcal{S}} - \mathbf{f}_2^{\mathcal{S}}\right\|_2 \leq \frac{1}{\sqrt{1 - \delta_k}} \left\|\mathbf{y}_1^{\mathcal{S}} - \mathbf{y}_2^{\mathcal{S}}\right\|. \tag{51}$$

Therefore, Equations (47), (50) and (51) give

$$\|\mathbf{f}_1 - \mathbf{f}_2\| \leq \left\|\mathbf{f}_1 - \mathbf{f}_1^{\mathcal{S}}\right\|_2 + \left\|\Phi^+\right\|_2 \left\|\mathbf{c}_2 - \mathbf{c}_2^{\mathcal{S}}\right\|_2$$
$$+ \frac{1}{\sqrt{1 - \delta_k}} \Big[ \|H\|_2 \left\|\mathbf{f}_1 - \mathbf{f}_1^{\mathcal{S}}\right\|_2$$
$$+ \left\|H\Phi^+\right\|_2 \left\|\mathbf{c}_2 - \mathbf{c}_2^{\mathcal{S}}\right\|_2 + \left\|\mathbf{y}_1 - \mathbf{y}_2\right\|_2 \Big]. \tag{52}$$

$$\leq \alpha \Big( \left\|\mathbf{f}_1 - \mathbf{f}_1^{\mathcal{S}}\right\|_2 + \left\|\Phi^+\right\|_2 \left\|\mathbf{c}_2 - \mathbf{c}_2^{\mathcal{S}}\right\|_2 \Big)$$
$$+ \frac{1}{\sqrt{1 - \delta_k}} \left\|\mathbf{y}_1 - \mathbf{y}_2\right\|_2, \tag{53}$$

$$\text{where } \alpha = 1 + \frac{1}{\sqrt{1 - \delta_k}} \|H\|_2. \tag{54}$$

Now, let $B = \Gamma(q_{\mathbf{f}}, p_{\hat{\theta}})$, i.e. the set of all joint distributions $\beta : \mathbb{R}^{n \times n} \to \mathbb{R}$ that have marginals $q_{\mathbf{f}}$ and $p_{\hat{\theta}}$. Also, let $\rho_{\hat{\theta}}$ be the distribution of $\mathbf{y} = H\mathbf{f}$ for $\mathbf{f} \sim p_{\hat{\theta}}$, i.e. the noiseless version of $\psi_{\hat{\theta}}$. Therefore, for $\beta \in B$,

$$\mathbb{E}_{(\mathbf{f}_1, \mathbf{f}_2) \sim \beta} \|\mathbf{f}_1 - \mathbf{f}_2\|_2 \leq \alpha \left[ \mathbb{E}_{\mathbf{f}_1 \sim q_{\mathbf{f}}} \|\mathbf{f}_1 - \mathbf{f}_1^{\mathcal{S}}\|_2 + \|\Phi^+\|_2 \mathbb{E}_{\mathbf{f}_2 \sim p_{\hat{\theta}}} \|\Phi\mathbf{f}_2 - \Phi\mathbf{f}_2^{\mathcal{S}}\|_2 \right]$$
$$+ \frac{1}{\sqrt{1 - \delta_k}} \mathbb{E}_{(\mathbf{f}_1, \mathbf{f}_2) \sim \beta} \|H\mathbf{f}_1 - H\mathbf{f}_2\|_2. \tag{55}$$

From Lemma A.1, we have

$$\mathbb{E}_{\mathbf{f}_1 \sim q_{\mathbf{f}}} \|\mathbf{f}_1 - \mathbf{f}_1^{\mathcal{S}}\|_2 = W_1(q_{\mathbf{f}} \| q_{\mathbf{f}}^{\mathcal{S}_k}) \leq \epsilon'. \tag{56}$$

Also, from Eq. (6),

$$\mathbb{E}_{\mathbf{f}_2 \sim p_{\hat{\theta}}} \|\Phi\mathbf{f}_2 - \Phi\mathbf{f}_2^{\mathcal{S}}\|_2 = \mathbb{E}_{\mathbf{g} \sim q_{\mathbf{g}}} \mathbb{E}_{\mathbf{f}_2 \sim p_{\hat{\theta}}(\cdot | \mathbf{g})} \|\Phi\mathbf{f}_2 - \Phi\mathbf{f}_2^{\mathcal{S}}\|_2, \tag{57}$$
$$= \mathbb{E}_{\mathbf{g} \sim q_{\mathbf{g}}} \mathbb{E}_{\mathbf{f} \sim p_{\hat{\phi}}(\cdot | \mathbf{g})} \|\Phi\mathbf{f} - \Phi\mathrm{proj}_{\mathcal{S}_k}(\mathbf{f})\|_2, \tag{58}$$
$$\leq \mathbb{E}_{\mathbf{g} \sim q_{\mathbf{g}}} \mathbb{E}_{\mathbf{f} \sim p_{\hat{\phi}}(\cdot | \mathbf{g})} \|\Phi\mathbf{f} - \Phi\mathrm{proj}_{\mathcal{S}_k}(\mathbf{f})\|_1, \tag{59}$$
$$\leq \epsilon. \tag{60}$$

Taking the infimum of Eq. (55) over $\beta \in B$, we get

$$\inf_{\beta \in B} \mathbb{E}_{(\mathbf{f}_1, \mathbf{f}_2) \sim \beta} \|\mathbf{f}_1 - \mathbf{f}_2\|_2 \leq \alpha(\epsilon' + \|\Phi^+\|_2 \epsilon) + \frac{1}{\sqrt{1 - \delta_k}} \inf_{\beta \in B} \mathbb{E}_{(\mathbf{f}_1, \mathbf{f}_2) \sim \beta} \|H\mathbf{f}_1 - H\mathbf{f}_2\|_2. \tag{61}$$

Note that the left-hand side of the above equation is precisely $W_1(p_{\hat{\theta}} \| q_{\mathbf{f}})$. Also, note that the rightmost term in Eq. (61) is $W_1(q_{\mathbf{y}} \| \rho_{\hat{\theta}})$:

$$W_1(q_{\mathbf{y}} \| \rho_{\hat{\theta}}) = \inf_{\beta \in B} \mathbb{E}_{(\mathbf{f}_1, \mathbf{f}_2) \sim \beta} \|H\mathbf{f}_1 - H\mathbf{f}_2\|_2. \tag{62}$$

From Eq. (44), since $q_{\mathbf{g}} = \psi_{\hat{\theta}}$, Lemma 3.1 implies $q_{\mathbf{y}} = \rho_{\hat{\theta}}$

$$\Rightarrow W_1(q_{\mathbf{y}} \| \rho_{\hat{\theta}}) = 0. \tag{63}$$

Combining with Eq. (61), and setting $\|\Phi^+\| \leq 1$ and $\alpha$ according to Eq. (54), we get

$$W_1(p_{\hat{\theta}} \| q_{\mathbf{f}}) \leq \left( 1 + \frac{1}{\sqrt{1 - \delta_k}} \|H\|_2 \right) (\epsilon + \epsilon') \tag{64}$$

$\square$

# B  Additional numerical experiments

## B.1  Evaluation of the posterior network

In order to evaluate the posterior network $h_\phi$, the following experiments were designed using the setup of the stylized MRI study. First, the consistency of the modeled posterior $p_\phi(\cdot | \mathbf{g})$ and the posterior induced by the main flow model $p_\theta(\cdot | \mathbf{g}) \propto q_{\mathbf{n}}(\mathbf{g} - H\mathbf{f}) p_\theta(\mathbf{f})$ was examined. For a fixed measurement vector $\mathbf{g}$, 50 images were sampled from $p_\phi(\cdot | \mathbf{g})$ using $h_\phi$. For the 50 samples, $\log p_\phi(\mathbf{f} | \mathbf{g})$ and $\log q_{\mathbf{n}}(\mathbf{g} - H\mathbf{f}) + \log p_\theta(\mathbf{f})$ were computed and plotted against each other. This was repeated was different measurement vectors $\mathbf{g}$. A scatter plot of $\log p_\phi(\mathbf{f} | \mathbf{g})$ versus $\log q_{\mathbf{n}}(\mathbf{g} - H\mathbf{f}) + \log p_\theta(\mathbf{f})$ is shown in Fig. 8 for the AmbientFlow trained on two different measurement configurations – (1) fully sampled noisy measurements, and (2) 4x undersampled noisy measurements. The scatter plots for each individual $\mathbf{g}$ are plotted in different colors. It can be seen that the slope of a linear fit of the scattered points is close to 1 for all three measurement vectors considered. This indicates that $p_\phi(\cdot | \mathbf{g}) \propto q_{\mathbf{n}}(\mathbf{g} - H\mathbf{f}) p_\theta(\mathbf{f})$, i.e. the modeled posterior and the posterior induced by the learned prior are consistent.

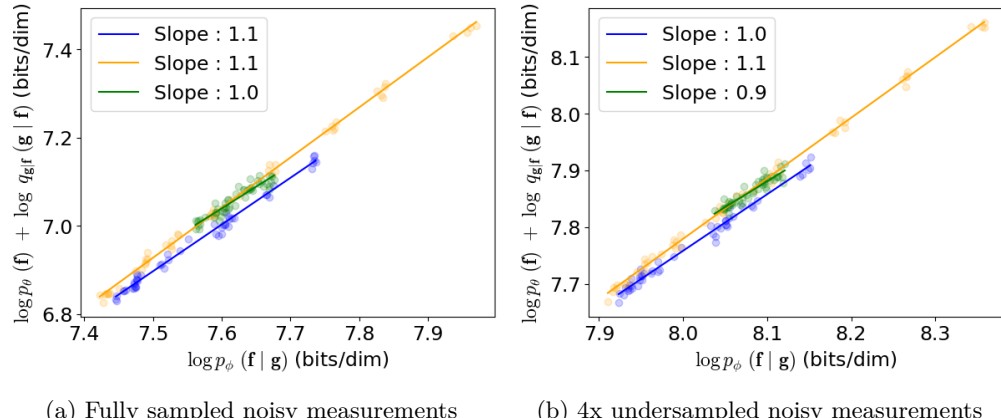

(a) Fully sampled noisy measurements

(b) 4x undersampled noisy measurements

Figure 8: Scatter plot of $\log p_\phi(\mathbf{f}\,|\,\mathbf{g})$ versus $\log q_{\mathbf{n}}(\mathbf{g}-H\mathbf{f})+\log p_\theta(\mathbf{f})$ for three different measurement vectors $\mathbf{g}$, for AmbientFlow trained using two different measurement configurations from the stylized MRI study. The scatter plots for each individual $\mathbf{g}$ are plotted in different colors.

Table 2: The mean $\pm$ standard deviation of the RMSE and SSIM values computed over the following test image datasets – (1) the MMSE estimates using the flow prior trained on true objects, and (3) the MMSE estimates using the AmbientFlow prior trained on measurements, and (3) the MMSE estimates using samples from $h_\phi(\cdot\,|\,\mathbf{g})$. The forward and noise models in the image reconstruction task were the same as the ones used to train the AmbientFlow.

| Method | $n/m = 1$ | | $n/m = 4$ | |
|---|---|---|---|---|
| | RMSE | SSIM | RMSE | SSIM |
| Flow prior trained on true objects | $0.017 \pm 0.002$ | $0.95 \pm 0.01$ | $0.022 \pm 0.003$ | $0.94 \pm 0.01$ |
| AmbientFlow prior | $0.016 \pm 0.002$ | $0.96 \pm 0.01$ | $0.025 \pm 0.004$ | $0.92 \pm 0.01$ |
| Posterior network $h_\phi(\cdot\,;\,\mathbf{g})$ | $0.016 \pm 0.002$ | $0.96 \pm 0.01$ | $0.026 \pm 0.004$ | $0.91 \pm 0.01$ |

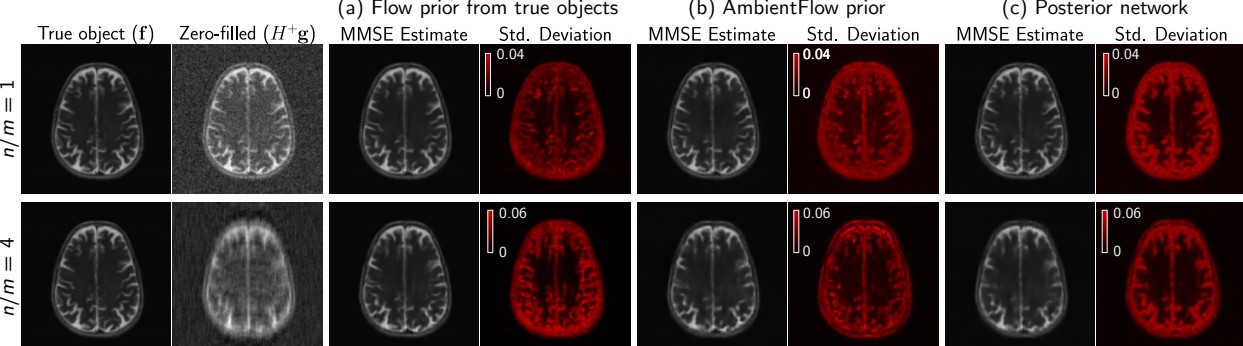

Figure 9: MMSE estimates obtained by an empirical average over images obtained from (a) Langevin dynamics-based posterior sampling that employs the flow prior trained on true objects, (b) Langevin dynamics-based posterior sampling that employs the AmbientFlow prior, and (c) the posterior network $h_\phi(\cdot\,;\,\mathbf{g})$.

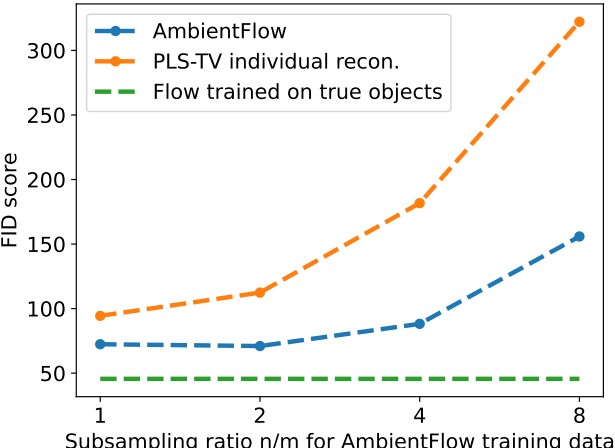

Figure 10: FID score as a function of the undersampling ratio used to simulate the training data for AmbientFlow for the stylized MRI study.

Table 3: MSE error in the mean of the images generated by the unconditional model relative to the true mean image; and relative squared-Frobenius error in the rank-1000 approximation $\Sigma_p$ of the learned covariance matrix relative to the rank-1000 approximation $\Sigma_q$ of the true covariance matrix.

|  | Conventional flow | $n/m = 1$ | $n/m = 2$ | $n/m = 4$ | $n/m = 8$ |
|---|---|---|---|---|---|
| $\|\mathbb{E}_{p_\theta}\mathbf{f} - \mathbb{E}_{q_\mathbf{f}}\mathbf{f}\|_2^2 / \|\mathbb{E}_{q_\mathbf{f}}\mathbf{f}\|_2^2$ | 0.3% | 0.8% | 0.7% | 0.4% | 1.4% |
| $\|\Sigma_p - \Sigma_q\|_F^2 / \|\Sigma_q\|_F^2$ | 14% | 16% | 15% | 15% | 35% |

Next, for the two measurement configurations within the stylized MRI study and for a test dataset of size 20, the outputs of the posterior model $h_\phi(\cdot\,;\mathbf{g})$ for a measurement input $\mathbf{g}$ was compared with the images reconstructed using the corresponding flow prior trained on the ground truths, as well as those reconstructed using the corresponding AmbientFlow prior. The empirical MMSE estimates as well as pixelwise standard deviation maps obtained from these posterior samples are shown in Fig. 9. The RMSE and SSIM of the MMSE estimates are shown in Table 2. It can be seen that the MMSE estimates computed using samples from $h_\phi(\cdot\,;\mathbf{g})$ closely resemble those computed via Langevin dynamics-based posterior sampling employing the AmbientFlow prior, both visually, as well as in terms of the RMSE and SSIM.

## B.2 Additional evaluation of unconditional models

In this section, we present additional evaluation and ablation studies on the learned unconditional models for the stylized MRI study. Figure 10 shows the FID score as a function of the undersampling ratio used to simulate the incomplete, noisy stylized MRI measurements used to train AmbientFlow. Next, we evaluate the ability of our models to learn first- and second-order image statistics. For this experiment, a conventional flow model trained on the true objects, as well as AmbientFlow models trained on noisy, undersampled stylized MRI measurements with $n/m = 1, 2, 4, 8$ were considered. Figure 11 shows the empirical mean, autocorrelation, and radial profile of the autocorrelation of images generated from the true unconditional distribution $q_\mathbf{f}$, and learned unconditional distribution $p_\theta$ for each of the flow models trained. The MSE error in the empirical mean image relative to the squared $\ell_2$ norm of the empirical mean of the true distribution is shown in Table 3. Next, a rank-1000 approximation of the sample covariance matrix was computed for all the models using randomized SVD of the data. Since the first and the 1000th singular values of this matrix differed by almost 6 orders of magnitude, a low-rank approximation to the covariance matrix differs minimally from the full covariance matrix in terms of the Frobenius norm, while also ensuring numerical stability in computation. The error in the empirical mean image relative to the squared $\ell_2$ norm of the empirical mean of

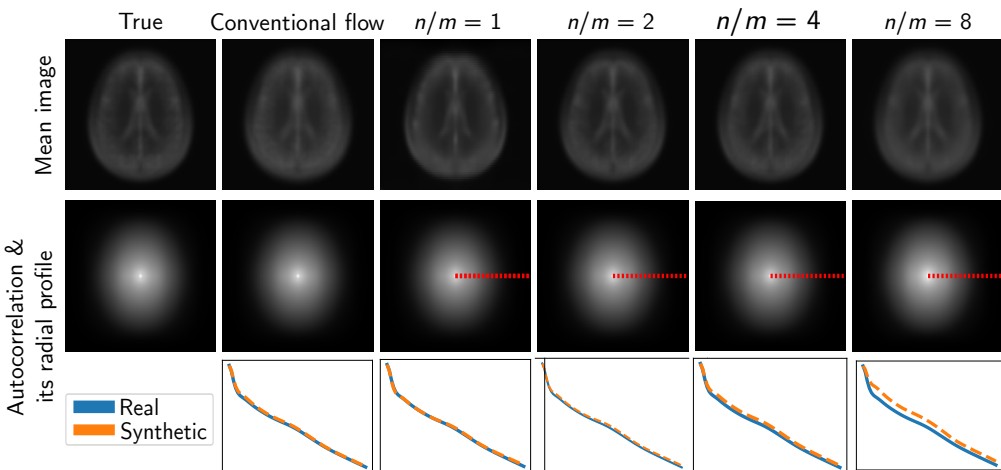

Figure 11: The first row shows the empirical mean of the images generated from the true unconditional distribution $q_{\mathbf{f}}$, and learned unconditional distribution $p_\theta$ for four different undersampling ratios considered in the sylized MRI study. The second row shows the image autocorrelations, and the third row shows the radial profile of the autocorrelation along the dotted red line.

Table 4: The mean ± standard deviation of the RMSE and SSIM values for the CelebA face image inpainting study computed over the following test image datasets – (1) the images reconstructed using the PLS-TV method, (2) the MAP and MMSE estimates using the flow prior trained on true objects, and (3) the MAP and MMSE estimates using the AmbientFlow prior trained on fully sampled noisy measurements.

| Method | | RMSE | SSIM |
|---|---|---|---|
| Navier-Stokes-based inpainting (Bertalmio *et al.*, 2001) | | $24.3 \pm 6.0$ | $0.81 \pm 0.03$ |
| Flow prior trained on true objects | MAP Estimate | $17.8 \pm 4.5$ | $0.81 \pm 0.05$ |
| | MMSE Estimate | $\mathbf{15.1 \pm 3.7}$ | $\mathbf{0.88 \pm 0.03}$ |
| AmbientFlow prior | MAP Estimate | $17.7 \pm 4.4$ | $0.80 \pm 0.03$ |
| | MMSE Estimate | $\mathbf{15.0 \pm 4.2}$ | $\mathbf{0.88 \pm 0.03}$ |

the true distribution is shown in Table 3. The squared-Frobenius error between the rank-1000 approximation $\Sigma_p$ and the learned covariance matrix relative to the rank-1000 approximation $\Sigma_q$ of the true covariance matrix relative to $\|\Sigma_q\|_F^2$ is shown in Table 3.

### B.3 Additional case study: face image inpainting using AmbientFlow prior

Here, the results of a case study of face image inpainting is presented. In particular, we consider the case where a trained uncomditional AmbientFlow is used as a prior in an image reconstruction task, where the forward operator applicable to the task is different from the forward operator used to train the AmbientFlow. The following two image reconstruction tasks are considered – (1) approximate maximum a posteriori (MAP) estimation, i.e. approximating the mode of the posterior $p_\theta(\cdot \,|\, \mathbf{g})$, and (2) approximate sampling from the posterior $p_\theta(\cdot \,|\, \mathbf{g})$.

For both the tasks, an AmbientFlow trained on noisy face images from the CelebA dataset, as well as a conventional flow model trained on the uncorrupted CelebA images were considered. For a held-out dataset of size 20, the first task was performed using the compressed sensing using generative models (CSGM) formalism (Asim *et al.*, 2020). For the second task, approximate posterior sampling was performed with the flow models as priors using annealed Langevin dynamics (ALD) iterations proposed in Jalal, *et al.* (Jalal *et al.*, 2021a).

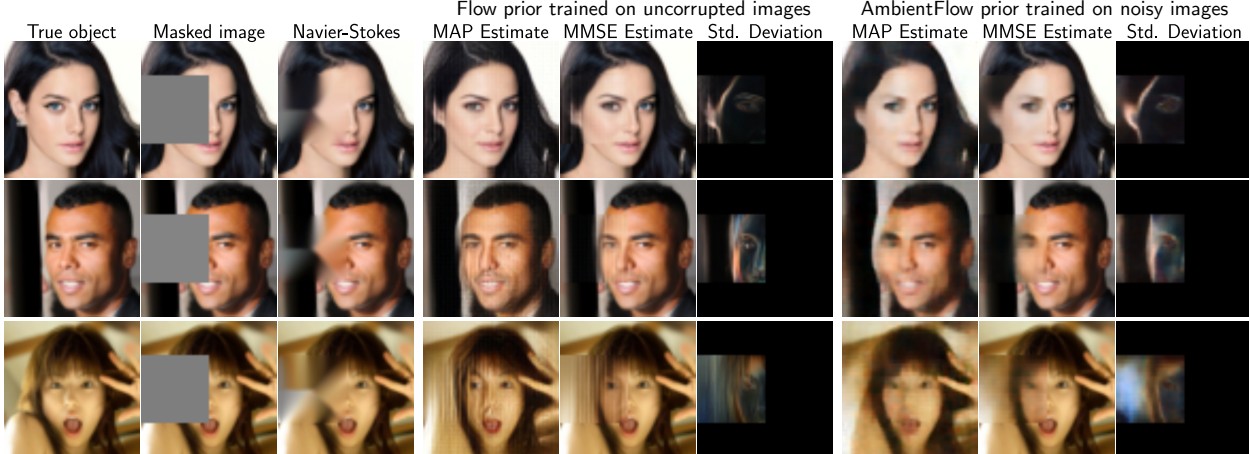

Figure 12: Image estimates and pixelwise standard deviation maps from the image inpainting case study.

For each true object, the MMSE estimate and the pixelwise standard deviation map $\hat{\sigma}$ were computed empirically using 36 samples obtained via ALD iterations. A baseline Navier-Stokes-based inpainting method was also compared (Bertalmio *et al.*, 2001). Table 4 shows the RMSE and SSIM of the reconstructed images with respect to the true object, averaged over a dataset of 20 test images. It can be seen that in terms of RMSE and SSIM, both the MAP and the MMSE image estimates obtained using the AmbientFlow prior are comparable to those obtained using the flow prior trained on uncorrupted images, despite the AmbientFlow being trained only using noisy CelebA images. Figure 12 shows the true and masked images along with the images inpainted by the algorithms described above. It can be seen that although the AmbientFlow prior provides comparable RMSE and SSIM estimates to the flow prior trained on uncorrupted images, it still retains some smoothing artifacts characteristic of the sparsity-promoting penalty imposed on it during training.

