# OpenReview forum: "AmbientFlow: Invertible generative models from incomplete, noisy measurements"
_TMLR — Accepted by TMLR_

### Review · Reviewer_gPnD · 2023-09-28

**Summary Of Contributions:**

The paper's goal is to recover a data distribution where observations are not direct samples from the distribution, but are rather measured through a known linear forward model with additive noise sampled from a known noise distribution.

The authors propose to tackle this problem in a probabilistic framework. Specifically, they build an unconditional generator as well as a measurement-conditional posterior network, both parameterized as an invertible neural network. They then leverage the known forward model and noise distribution to train the posterior network to inver the forward model and generate "clean" data samples $\mathbf{f}$ from noisy measurements $\mathbf{g}$. They then train the generator to match the distribution of the posterior network.

The authors now prove that under certain assumptions that are of practical interest, optimizing their objective minimizes the KL-divergence of $p_\theta$ and the underlying, true data distribution, thus successfully learning a generative model of the true data distribution.

**Audience:**

Yes

**Broader Impact Concerns:**

-

**Claims And Evidence:**

Yes

**Requested Changes:**

Minor:
- I have never heard "computed imaging" before. We usually call it "Computational Imaging", see wikipedia article [here](https://en.wikipedia.org/wiki/Computational_imaging)
- "that is represents" --> "that represents" in paragraph 3 of section 3.
- Wienier --> Wiener in Baselines and evaluation metrics

**Strengths And Weaknesses:**

# Strengths
- The paper is very well written. Notation is clear and concise. The level of abstraction is appropriate.
- The paper has both theoretical and numerical results.
- The numerical results are presented well.

# Weaknesses
- The conditional posterior was not evaluated, and hence, the comparisons in the paper are a bit confusing to me. In Fig. 4, for instance, the authors show qualitative results of applying wiener deconvolution and BM3D to measurements. However, the appropriate comparison for BM3D and Wiener deconvolution would be the output of the trained posterior on the same images, or at least a reconstruction algorithm that uses the unconditional prior. The unconditional samples do not allow for a principled qualitative comparison.
- The authors only evaluate the unconditional generator numerically, not the conditional posterior.
- The authors claim "An estimate of the distribution of objects is known to be important for applications in imaging science.". I agree, but I think the authors would do well to motivate why exactly that is the case. Specifically, the question the authors should answer: why do we care about the unconditional model when we perform reconstruction? The only reason that I can see is that the measurement forward model is *different* at test time than it was at training time, such that the posterior network would be out-of-distribution. In this case, the unconditional prior enables us to solve the inverse problem still.
  - I would hence expect that the authors evaluate exactly this case. I.e., run an experiment where you show that you can reconstruct measurements subject to a forward model that was *different* than the one the model was trained on. For instance, a different class of blur kernels, or different noise levels.
  - This is also extremely important b/c otherwise, I would expect the authors to benchmark with a method that is trained to directly invert the measurement model, i.e., a state-of-the-art image-to-image translation model that is trained to directly reconstruct a noisy & blurry image.

---

> ### Author Response · Authors · 2023-11-02
> **Thank you for the useful comments, we have now performed several experiments to further evaluate the posterior**
>
> - **Regarding comparisons in Fig. 4**: The purpose of the comparisons shown in Fig. 4 is not to demonstrate the performance of AmbientFlow within the context of a task, such as image reconstruction. Rather, its purpose is to illustrate how well the AmbientFlow learns the underlying unconditional distribution of objects. The comparison with images individually reconstructed from the measurements using classical reconstruction approaches such as BM3D, Wiener deconvolution or PLS-TV was provided to demonstrate that these individual reconstructed images are not an acceptable proxy to the objects for the purposes of training a generative model of objects. This suggests that an AmbientFlow directly on the measurements would give a better approximation to the object distribution as compared to a regular flow model trained on these individually reconstructed image datasets.
>
> - **Posterior model evaluation**: We thank the reviewer for pointing this out. We have now performed additional numerical experiments that evaluate the outputs of the posterior network for the stylized MRI study. First, the consistency of modeled posterior $p_\phi(\mathbf{f} | \mathbf{g})$ and the posterior induced by the learned prior $p_\theta(\mathbf{f} | \mathbf{g}) \propto q(\mathbf{g} | \mathbf{f}) p_\theta(\mathbf{f})$ was examined. Additional details about this study are provided in the response to reviewer JKdo.
>
>     Next, for the two measurement configurations within the stylized MRI study, the outputs of the posterior model for a particular measurement input g was compared with the image reconstructed from g using a non-data-driven reconstruction algorithm, those reconstructed using the corresponding flow prior trained on the ground truths, as well as those reconstructed using the corresponding AmbientFlow prior. We observed that in terms of RMSE, the posterior model performed similar to an image reconstruction method which employed the corresponding main flow model as the prior. We will add these results to the appendix of the updated manuscript.
>
> - **Motivating the need for unconditional generative models**: We thank the reviewer for pointing out the need to better motivate unconditional generative models. The line in question will be replaced with the following lines in the discussion:
>
>     “An estimate of the distribution of objects is known to be important for applications in imaging science. This is because an unconditional generative model of objects can potentially be used in image reconstruction tasks without the need for paired data of images and measurements, and in a way that accommodates a wide variety of relevant forward models [1,2]. Additionally, unconditional generative models can be used to approximate ideal Bayesian classifiers [3], or for anomaly detection [4], or image manipulation [5]”
>     1. Bora, Ashish, et al. "Compressed sensing using generative models." ICML. PMLR, 2017.
>     2. Asim, Muhammad, et al. "Invertible generative models for inverse problems: mitigating representation error and dataset bias." ICML. PMLR, 2020.
>     3. Zhou, Weimin, et al. "Ideal Observer Computation by Use of Markov-Chain Monte Carlo with Generative Adversarial Networks." arXiv:2304.00433 (2023).
>     4. Zhao, Yuzhong, et al. "AE-FLOW: Autoencoders with Normalizing Flows for Medical Images Anomaly Detection." The Eleventh ICLR. 2022.
>     5. Torres, Ricardo da S., and Ulf Brefeld. "Principled Interpolation in Normalizing Flows."
>
> - **Image reconstruction using a generative prior with test-time forward model shift**: With respect to the reviewer’s question about using the learned prior to solve an inverse problem in the case where the forward model is different from the one associated with the training measurements, this has already been presented for the stylized MRI study. Specifically, in Fig. 7 & Table 1, we show the results of image reconstruction from 4x undersampled noisy MRI measurements, where the AmbientFlow prior was trained using fully sampled noisy measurements. However, we agree that we had not adequately emphasized the test-time shift in the forward model. This will be fixed in the updated version, which will also contain another case study for the CelebA dataset, where the AmbientFlow trained on noisy face images was used as a prior in an inpainting task.
>
> - **"Computed imaging" vs. "computational imaging"**: The term "computed imaging" has been frequently used in the literature at least since the 1980s [1]. We agree with the reviewer that "computational imaging" has also often been used with the same meaning. However, we have observed that it has also been used sometimes to refer to any sort of computation applied to images [2]. Therefore, we prefer to keep the more precise term "computed imaging".
>     1. Hounsfield, Godfrey N. "Computed medical imaging." Science 210.4465 (1980): 22-28.
>     2. Eils, Roland, and Chaitanya Athale. "Computational imaging in cell biology." The Journal of cell bio. 161.3 (2003): 477-481.

---

### Review · Reviewer_8cDe · 2023-10-11

**Summary Of Contributions:**

The authors present a framework for learning flow models from linearly corrupted, noisy data. The problem of learning from corrupted data is important for scientific applications where access to clean data is expensive or impossible. Prior to this work, the problem of learning generative models from corrupted data has been explored in the context of GANs and diffusion models. The authors extend this line of work to Flow Models that have the advantage that model the likelihood function directly. The proposed method is based on a novel training objective that uses two networks: a model that generates clean samples conditioned on measurements and a model that gets the generated samples and maximizes their likelihood. The authors demonstrate the empirical performance of their method in toy datasets, CelebA-64x64 and a stylized MRI application.

**Audience:**

Yes

**Broader Impact Concerns:**

I do not have broader impact concerns regarding this paper.

**Claims And Evidence:**

Yes

**Requested Changes:**

I think the paper would improve by strengthening the experimental evaluation. It would be nice to compare with Ambient GAN / Diffusion on Celeb-A 64x64 or some other image dataset. It would also be nice to include plots of how the performance (in FID) changes as we increase the noise or the blurring. I would also encourage the authors to experiment with different types of linear corruptions, e.g. random inpainting or block inpainting as in prior work.

**Strengths And Weaknesses:**

I believe this is a strong submission and of interest to the TMLR community.

Strengths:

- Importance of the problem: learning from corrupted data is a very interesting research direction with many scientific applications.
- Relevance: this is the first framework for learning generative models from corrupted data. It extends a line of recent works on learning generative models from corrupted data and it proposes the first solution for learning flow models from measurements.
- Presentation: The paper is clearly written and easy to follow.
- Novelty of the method: the proposed method is interesting. The authors circumvent the difficulty of modeling the likelihood of fake measurements by introducing an auxiliary network that produces clean samples from the posterior distribution given the lossy observations. The objective penalizes the auxiliary network if the measured generations drift away from the distribution of observed measurements. The authors use the fact that for certain corruptions, if the measured distributions match, then it must be that the distributions before the measurements match. This ensures that the auxiliary network produces samples from the right distribution and with those, Flow Models can be trained to model the true likelihood.
- Correctness and novelty of theoretical results: I read the theoretical results and proofs of the paper and seem novel and correct to me. The authors establish the correctness of the training objective, analyze cases in which you can learn perfectly from measurements and finally obtain approximate recovery guarantees in the case of matrices that have a null-space.


Weaknesses:

- I think the condition that H is full-rank in Lemma 3.1 can be relaxed in certain cases and still get optimal reconstruction results. I refer the authors to results from the Ambient GAN paper, e.g. for the random inpainting case.
-  The experimental evaluation is not very thorough. Comparisons with baselines are missing. Ablation studies regarding how performance changes with increased corruption are limited.
- The authors missed some recent work in learning diffusion models from corrupted data. I refer the authors to the following works:
Solving Inverse Problems with Score-Based Generative Priors Learned from Noisy Data, Ambient Diffusion: Learning Clean Distributions from Corrupted Data.
- The proposed method bears some resemblances to AmbientGAN. Particularly, both works look at how far measured generated images are from the real measurements. This is the critical component for the proposed objective to work and I think a more thorough discussion of the similarities and differences to AmbientGAN would have been helpful.

---

> ### Author Response · Authors · 2023-11-02
> **Thank you for your supportive comments, please see a detailed response below.**
>
> - **Extending Lemma 3.1 beyond full-rank forward operators**: We agree that Lemma 3.1 can be relaxed to include a larger family of forward operators. Specifically, it can be extended to any measurement model for which the push-forward operator mapping the object and measurement distributions is invertible. However, forward models such as a random inpainting model that may lead to an invertible push forward operator are not necessarily relevant for practical computed imaging systems. This type of a measurement model when deliberately introduced could be interesting for learning a generative model that avoids memorization [1], but designing a numerical study that systematically investigates this phenomenon would be a separate and distinct endeavor. Moreover, deterministic forward models with a fixed null space are quite relevant in computed imaging, where it is typically not possible to easily control the imaging operator in a way that guarantees an invertible push-forward operator. Several works talk about the difficulty of realizing such random acquisition patterns and the inevitability of a fixed null space [2,3]:
>     1. Daras, Giannis, et al. "Ambient Diffusion: Learning Clean Distributions from Corrupted Data." arXiv preprint arXiv:2305.19256 (2023).
>     2. Graff, Christian G., and Emil Y. Sidky. "Compressive sensing in medical imaging." Applied optics 54.8 (2015): C23-C44.
>     3. Pauly, John M. "Compressed sensing MRI." Signal Processing Magazine, IEEE 25.2 (2008): 72-82.
> However, in the updated manuscript to be uploaded soon, we have now mentioned the possible relaxation of Lemma 3.1 after its statement, and have referred the reader to the AmbientGAN paper for additional information.
>
> - **Additional numerical experiments**: We have now expanded the numerical evaluation by adding additional experiments evaluating the posterior models. Additionally, we are also in the process of adding further ablation studies where the performance of the AmbientFlow as a function of undersampling rate is studied for the stylized MRI study. We will also be adding a case study on using AmbientFlow trained on noisy face images as a generative prior for a face inpainting task. We are close to finishing these experiments, and will be updating the manuscript soon.
>
> - **Missing references on diffusion models from corrupted data**: We thank the reviewer for pointing us these two important works that advance diffusion models in the ambient setting; we will be acknowledging and appropriately discussing them in the revised version.
>
> - **Discussion on similarities and differences with AmbientGAN**: We agree with the reviewer that a more thorough discussion of the similarities and differences between AmbientGAN and AmbientFlow is warranted. The two frameworks are similar in the sense that they both attempt to match the distribution of real and synthetic measurements as a proxy for matching the distribution of objects. The basic differences between GANs and Flows in the context of downstream Bayesian inference tasks in imaging have already been discussed in the manuscript, and these differences also apply to AmbientGAN and AmbientFlow. Further similarities and differences between the two are as follows. In some scenarios, AmbientGAN can be more flexible since it can easily accommodate arbitrary differentiable random forward models, whereas AmbientFlow is limited by the capabilities of the posterior network. However, in the presence of a fixed null space, some studies have shown that AmbientGAN performance can degrade and the images produced by it can display aliasing artifacts characteristic of the measurement operator [REF]. To the best of our knowledge, incorporating additional priors into the AmbientGAN loss function has not been rigorously studied. Another key difference between the two approaches is that unlike AmbientGAN, AmbientFlow also provides a posterior network which can be useful by itself for certain image reconstruction tasks.

---

> > ### Author Response · Authors · 2023-11-10
> > **Response continued**
> >
> > - **Numerical comparison with AmbientGAN**: We believe that it is fundamentally difficult to set up a meaningful comparison between AmbientGAN and AmbientFlow, since they are frameworks applicable to two different classes of generative models, rather than being specific generative model architectures themselves. It is possible that for some imaging models, AmbientGAN employing a specific GAN architecture such as the StyleGAN2 produces images with different FID than AmbientFlow employing a specific architecture such as Glow. However, a large confounding factor in such a comparison is the inherent difference in performance between StyleGAN2 and Glow architectures, and there is no way to cast StyleGAN2 in the variational Bayesian framework of AmbientFlow in order to compare the two frameworks fairly. This is why we refrain from thinking of AmbientGAN as a necessary baseline. Given the limited amount of time, we think that other experiments such as further ablation studies and posterior evaluation would provide more value to the paper.

---

### Review · Reviewer_JKdo · 2023-10-18

**Summary Of Contributions:**

The authors propose a normalizing flow-based framework to train a generative model for a distribution observed through a known linear  operator and corrupted with known noise. The framework resembles AmbientGAN but adapts it to the setting of flows trained by maximum likelihood. This adaptation entails training a separate posterior network which is a conditional normalizing flow (and which is useful in its own right) and a bespoke loss to facilitate the minimization of the Kullback–Leibler divergence between the measurement distribution and the variational approximation. The authors provide an analysis of the distributional error for low-dimensional (k-sparse) objects under idealized training and apply their framework to MAP estimation and posterior sampling in several stylized problems.

**Audience:**

Yes

**Broader Impact Concerns:**

--

**Claims And Evidence:**

Yes

**Requested Changes:**

- The idea to train a posterior network reminds me a bit of what Sun and Bouman do in "Deep Probabilistic Imaging" (AAAI 2021) to sample from the posterior given a known forward model and a generative prior. It is not the same but the connection is worth mentioning.

- KL divergence makes sense for distributions with common support. For low-dimensional inputs (e.g. k-sparse), a perfectly trained model would have to become non-injective when the latent distribution is a white Gaussian, and in particular the volume change formulas would collapse. Does this cause any issues in theory or practice? It would be nice to see a comment.

- In AmbientGAN the forward operators can also be random. Although you do not do this, it seems that your framework should accommodate it (except for some parts of analysis). Perhaps a toy example along these lines would further strengthen the manuscript.

- projection on k-sparse signals looks like hard thresholding; how is this implemented in the learning pipeline?

- Please carefully check the writing ("is approximates the object", "remainder of the _chapter_", "is introduced that is represents", "Wienier deconvolution", ..., I would also suggest to replace "fake measurements" by something like "synthetic measurements" since at least to my ear fake has a pejorative connotation)

**Strengths And Weaknesses:**

### Strengths

This is a very nice paper which proposes a creative adaptation of AmbientGAN to normalizing flows. The structure and the prose is clear. The experiments convincingly illustrate the ideas and the empirics are supported by theoretical results. The addressed problem---estimating a distribution from indirect observations---is relevant in a number of domains.

### Weaknesses

- (minor) The number of training samples for a 2D toy problem in Figure X seems excessive. It would be interesting to investigate how the quality of the result scales with the amount of training data.

- It is unclear how well in practice the prior and the posterior are modeled. One way to check this would be to use images that are Gaussian random fields for which all the distributions can be explicitly characterized if the forward operator is linear.

---

> ### Author Response · Authors · 2023-11-02
> **Thank you for the encouraging review, please see the detailed point-by-point response below**
>
> - **Size of the training dataset in the toy problem**: The effective dataset size for this experiment seems excessive because for this particular experiment, the measurements were sampled on-the-fly from the true measurement distribution $q_{\mathbf{g}}$. We will add visualizations from intermediate training iterations that represent a lower effective dataset size.
>
> - **How well are the prior and the posterior modeled, and the use of Gaussian random fields**: Using Gaussian random fields in which the prior and the posterior can be theoretically computed is a good way to test how well the AmbientFlow learns multivariate Gaussian priors and the associated posteriors. However, multivariate Gaussian random fields are completely characterized by first and second order statistics. Therefore, such an experiment would still only test the ability of the model to learn first and second order statistics. The ability of our models to learn first and second order statistics can also be empirically tested with the existing models.
>
>     In order to test the fidelity of learned first and second-order statistics, we considered AmbientFlow trained via two stylized MRI experiments - (1) the experiment with fully sampled, noisy MRI measurements, and (2) the experiment with the 4x undersampled, noisy MRI measurements. For these two experiments, we observed that the MSE between the pixelwise means of the real and synthetic images was, respectively, 0.8% and 0.3% of the squared L2 norm of the mean of real images. In order to test the fidelity of the second-order statistics, we computed a rank 1000 approximation of the covariance matrices of real and synthetic images using randomized SVD of the data. We observed that the squared Frobenius norm of the difference between these two matrices was around 5% of the squared Frobenius norm of the real covariance matrix for both the experiments.
>
>     In order to better assess the learned posterior, the consistency of modeled posterior $p_\phi(\mathbf{f} | \mathbf{g})$ and the posterior induced by the learned prior $p_\theta(\mathbf{f} | \mathbf{g}) \propto q(\mathbf{g} | \mathbf{f}) p_\theta(\mathbf{f})$ was examined by sampling points from the modeled posterior for a fixed g, and comparing the values of $\log p_\phi(\mathbf{f} | \mathbf{g})$ and $\log q(\mathbf{g} | \mathbf{f}) + \log p_\theta(\mathbf{f})$. We observed that a scatter plot of $\log p_\phi(\mathbf{f} | \mathbf{g})$ versus $\log q(\mathbf{g} | \mathbf{f}) + \log p_\theta(\mathbf{f})$ closely matched a straight line, whose slope varied between 0.9 and 1.1 when different values of the measurement vector $\mathbf{g}$ were considered. This indicated that $\log p_\phi(\mathbf{f} | \mathbf{g})$ and $\log q(\mathbf{g} | \mathbf{f}) + \log p_\theta(\mathbf{f})$ approximately differ only by a constant, implying that $p_\phi(\mathbf{f} | \mathbf{g}) \propto q(\mathbf{g} | \mathbf{f}) p_\theta(\mathbf{f})$ as desired. We will be adding the scatter plots of $\log p_\phi(\mathbf{f} | \mathbf{g})$ versus $\log q(\mathbf{g} | \mathbf{f}) + \log p_\theta(\mathbf{f})$ to the updated manuscript soon.
>
> - **Connection to "Deep Probabilistic Imaging" by Sun and Bouman**:
>     Thank you for pointing this out. Although our work addresses a fundamentally different problem, there are indeed some interesting similarities between the posterior model in our work and the posterior model in Sun and Bouman’s work. Namely, both works use an invertible network to model the posterior, and train it in a way that is constrained by a prior. Sun and Bouman estimate the parameters of the posterior network at test time given a measurement vector by iteratively optimizing an objective constrained by a hand-crafted, sparsity-based prior. The posterior model in our work is a conditional generative model that, when trained, can directly produce posterior samples by taking in the measurement vector as one of the inputs, without the need for further optimization. We will add a note regarding this to the revised manuscript.

---

> ### Author Response · Authors · 2023-11-02
> **Response to initial review continued.**
>
> - **KL divergence and low-dimensional signals**: This is a very interesting comment. It is true that if the support of objects is only restricted to k-sparse vectors, the log-likelihood objective would be ill-defined. However, assuming that the true distributions are representable by INNs makes sure that the theory applies to only distributions with full support, since a distribution supported only on k-sparse vectors would not be representable by an INN. Note that we only require the true object distribution to be largely concentrated on the set of sparse vectors, and do not require it to be supported only on sparse vectors.
>
>     On the practical side, the objects of practical interest are not exactly sparse, but only approximately so; they are characterized by a few large coefficients with a large number of relatively small coefficients in a suitable transform domain. Accurately modeling such a distribution requires an INN architecture that allows for Jacobians with large condition numbers. Note that this applies equally to conventional flow models that are trained directly on natural/medical images, in addition to the proposed ambient models. Fortunately, in modern INNs, the trainable weights together model the log-determinant of the Jacobian directly. This enables the stable computation of log-determinant of Jacobians with potentially large condition numbers.
>
> - **Random forward operators**: In theory, our framework accommodates the case where the forward operator is random. Specifically, it is straightforward to modify the loss function so that in addition to $\mathbf{g}$, the random forward operator $H \sim q_H$ is also a conditioning input for the posterior model. However, in practice, we observed that designing a posterior network architecture that can successfully account for the random forward operator is nontrivial. We have mentioned this limitation in the discussion section of the manuscript. With this being said, deterministic forward operators with a fixed null space are still highly relevant for computed imaging. Designing a posterior that can take in the parameters of a random forward model as a conditioning input would be interesting to explore in the future.
>
> - **Implementation of the sparsity penalty**: The output of the posterior network is hard-thresholded after a sparsifying transform in order to obtain the projection onto the set of k-sparse signals, and the L1 norm of the residual is used as an additive penalty, with a regularization parameter μ. k and μ are treated as tunable hyperparameters. However, note that the loss terms for both the INNs correspond to the original (un-thresholded) outputs of the posterior, and the projection is used only to compute the additive penalty. This is consistent with the actual expression of the loss function in Eq. (10), and ensures that the loss terms from both the INNs are well-defined.
>
> - **Issues in the writing**: Thank you for pointing these out. We will fix these issues in the updated version. We agree that “synthetic images/measurements” is more appropriate.

---

> ### Comment · Reviewer_JKdo · 2023-11-10
>
> The following
>
> > However, multivariate Gaussian random fields are completely characterized by first and second order statistics. Therefore, such an experiment would still only test the ability of the model to learn first and second order statistics. The ability of our models to learn first and second order statistics can also be empirically tested with the existing models.
>
> is not completely true. Gaussians are indeed characterized by first and second order statistics but there are infinitely many distributions which have the same first and second order statistics as a given Gaussian so reproducing them does not mean much. The important part is that the higher order statistics are very specific functions of E X and E XX^T. The uniquely interesting aspect of Gaussians is that we have a closed form expression for the posterior with a linear forward operator and even looking at the samples (from the exact and approximate posteriors) should be illuminating. Is there a particular reason you think this experiment is not appropriate?

---

> ### Author Response · Authors · 2023-11-15
> **Response by authors**
>
> We agree with the reviewer that such an experiment would be informative. However, it would not be possible to perform these experiments before the reviewer recommendation deadline. In our revised manuscript uploaded to OpenReview, we do not make claims regarding the distributional accuracy of the models learned in our experiments that go beyond the scope of the considered evaluation metrics. Therefore, we believe that the proposed experiment is not critical to support the claims made. However, since this is an interesting experiment, we would be willing to perform it after the recommendation deadline if the paper is accepted.

---

### Decision · Action_Editor_MyJU · 2023-12-04

**Recommendation:** Accept as is

**Comment:**

The paper is clear, well structured, and support the claims that the authors made. The authors build on the previous AmbientGAN work to provide a substantial contribution, especially in the medical imaging domain.

**Audience:**

The paper will be of interest to both a probabilistic learning of distributions from corrupted data and medical imaging audiences.

**Claims And Evidence:**

The authors propose an approach to learn invertible generative models from corrupted data, a claim that is clearly explained and demonstrated both theoretically (under specific assumptions) and empirically in the medical imaging setting (MRI data).